# Envisaging the job satisfaction and turnover intention among the young workforce: Evidence from an emerging economy

Xuelin Chen[1,2], Abdullah Al Mamun [2]*, Wan Mohd Hirwani Wan Hussain[2], Gao Jingzu[2], Qing Yang[2], Sayed Samer Ali Al Shami[3]

1 School of Business, Jishou University, 416000, Jishou City, Hunan, China, 2 UKM—Graduate School of Business, Universiti Kebangsaan Malaysia, UKM Bangi, Kajang, Selangor Darul Ehsan, Malaysia, 3 Institute of Technology Management and Entrepreneurship, Universiti Teknikal Malaysia Melaka, Melaka, Malaysia

* almamun@ukm.edu.my/mamun7793@gmail.com

**Data Availability Statement:** All relevant data are within the manuscript and its Supporting Information files.

## Abstract

As the economy evolves and markets change after Covid-19, demand and competition in the labor market increase in China, and employees become increasingly concerned about their career opportunities, pay, and organizational commitment. This category of factors is often considered a key predictor of turnover intentions and job satisfaction, and it is important that companies and management have a good understanding of the factors that contribute to job satisfaction and turnover intentions. The purpose of this study was to investigate the factors that influence employees' job satisfaction and turnover intention and to examine the moderating role of employees' job autonomy. This cross-sectional study aimed to quantitatively assess the influence of perceived career development opportunity, perceived pay for performance, and affective organisational commitment on job satisfaction and turnover intention, as well as the moderating effect of job autonomy. An online survey, which involved 532 young workforce in China, was conducted. All data were subjected to partial least squares-structural equation modelling (PLS-SEM). The obtained results demonstrated the direct influence of perceived career development, perceived pay for performance, and affective organisational commitment on turnover intention. These three constructs were also found to have indirect influence on turnover intention through job satisfaction. Meanwhile, the moderating effect of job autonomy on the hypothesised relationships was not statistically significant. This study presented significant theoretical contributions on turnover intention in relation to the unique attributes of young workforce. The obtained findings may also benefit managers in their efforts of understanding the turnover intention of the workforce and promoting empowerment practices.

## Introduction

With the constant development of the economy and the increasing competition in the labour market, an increasing number of young employees are contemplating job-hopping or leaving their current company [1–3]. The higher the turnover intention, the higher the likelihood of

**Funding:** The author(s) received no specific funding for this work.

**Competing interests:** The authors have declared that no competing interests exist.

quitting the job [4]. Belete [5] described turnover intention as the intention to leave the workplace. The management must have good understanding of the factors that contribute to turnover intention considering how a higher turnover rate would affect the organisational morale, reduce the sense of identity within the company [6], and yield negative financial impact since the companies have to allocate resources for recruitment and training and experience brain drain and lower productivity [5]. Therefore, how to reduce the turnover intention among the young workforce and improve employee job satisfaction is an area that most organizations need to consider.

Currently, the young millennial workforce has become the largest group in the labor market, and how to retain these young employees has been an ongoing issue and challenge for most organizations and managers [7,8]. In the past, the older workforce possesses more traditional values, such as contributing to the country and supporting the family, but the younger workforce emphasises more worldly values and embraces self-development as the key purpose of working [9]. However, compared to middle-aged and older employees with more work experience and years of experience, the younger workforce focuses more on a number of unique attributes that in turn can have a significant impact on their job satisfaction and turnover intention, such as room for personal growth and development [10], pay level and task clarity [11], job variety and job autonomy [12]. Therefore, this study attempts to examine the mechanisms influencing young employees' job satisfaction and turnover intention, as well as the moderating role of job autonomy, by looking at some unique attributes, such as personal career development opportunities.

In addition, under the impact of Covid-19 on the economy and labor market, China's youth labor market is facing a serious social problem-high brain drain most young new employees will leave or jump jobs within 2–3 years, with most of them being fresh university graduates, which in turn brings some hidden problems such as social problems of jobless youth [3]. Therefore, improving job satisfaction and reducing the turnover rate of young employees during their working life is an urgent issue that most Chinese companies and organizations need to address. Due to China's unique family planning policy, most young employees are now as independent and capable as only children and are more focused on their own interests and development, which in turn leads to a stronger attachment to salary and development opportunities due to both financial and family pressures [3,12]. Therefore, understanding the perception and importance of salary and career development opportunities among young Chinese employees can effectively help companies and organizations find the right solution to retain this fresh and capable young workforce.

Furthermore, young workers in the workplace usually focus on things that are related to their own interests, such as career development opportunities and salary [8,13]. Young employees are ready to change jobs when the benefits and development space provided by the job do not meet their expectations and satisfy their needs [3]. At the same time, the imbalance between job income and personal living expenses will also put more stress on them, which in turn will lead to higher intention of young employees to leave their jobs [7]. In addition, affective commitment is based on the emotional bond that employees build with the organization through positive work experiences, and some young employees' lack of work experience and skills, among other factors, may lead them to be reluctant to spend too much time building a strong emotional bond with the organization, which in turn leads to the idea of low cost of leaving and dissatisfaction with the job status quo [14]. Therefore, this study wanted to explore the perceptions and the level of importance of career development opportunities, pay for performance, and affective organizational commitment for young workers, and then discuss in depth whether these factors have some impact on the turnover intention and job satisfaction of young workers.

From the theoretical viewpoints, this study provided better understanding on the relationships of PC and PP with TI of young employees. Secondly, this study provided valuable insights on the unique attributes of the young workforce in relation to TI. Meanwhile, from the practical viewpoints, the obtained findings of this study on factors that influence TI and promote higher work efficiency can substantially benefit companies and managers to attain improvement and success. Addressing significant factors that influence TI, instead of the actual turnover, provides valuable opportunities for the managers to implement potential remedial measures prior to the actual turnover.

## Literature review

### Theoretical foundation

The equity theory suggests that individuals are likely to compare their own inputs and corresponding outcomes with other individuals, resulting in the formation of their own perceptions of fairness [15]. Examples of inputs include loyalty, commitment, experience, skills, and flexibility, while examples of outcomes include remuneration, development, benefits, reputation, enjoyment, and rewards [16]. Individuals who experience unfairness at the workplace are more likely to be demotivated and eventually yield lower inputs or seek changes. On the other hand, objective evaluation of the work performance would ensure fairness and yield higher inputs; knowing that good performance is rewarded, employees would work harder to be rewarded [15]. The application of the equity theory to explore turnover is deemed fitting, as TI serves as the resultant outcome of inequity [16]. Creating a fair working environment is part of the management's measures to reduce TI among the employees.

Meanwhile the job demands-resources (JD-R) Model was first introduced by Demerouti et al. [17] and gained high popularity among researchers. The JD-R model posits that any job can be described by two sets of variables: job demands and job resources [18]. In particular, job resources are defined by Demerouti et al. [17] as "those physical, social, or organizational aspects of the job that may serve any of the following functions: (a) facilitate achievement of work goals, (b) reduce job demands and their associated physiological and psychological costs, and (c) promote personal growth and development." Hence, career development opportunities, pay for performance and affective organizational commitment can all be considered job resources, since job resources include any aspects of the job that can stimulate personal growth and development. Job resources can contribute to positive outcomes and also act as protective factors against negative outcomes, such as turnover intention [19]. In other words, employees who have access to more job resources are less likely to exhibit turnover intention.

### Development of hypotheses

**Perceived career development opportunity (PC).** PC refer to the employees' perceptions of the availability of work assignments and job opportunities that align with their career interests and goals within their current organization [20]. Training for career advancement and growth can create the sense of being valued and appreciated for employees [21]. Employees' JS can be enhanced through professional development opportunities and flexible working hours (with more scheduling options) [22]. This finding is corroborated by the JD-R model, which suggests that employees who have access to job resources tend to be more engaged and satisfied at work [19]. Meanwhile, Muleya et al. [23] have underscored the impact of career development opportunity, regarded as one key job resource, on employees' attitudes in the workplace. It is reasonable to assume that when employees are provided with career development opportunities, they are more likely to exhibit positive work attitudes. Additionally, Barhate and Dirani [24] posit that the younger generation of workers require immediate rewards

in the form of promotions and career advancement opportunities for a job well done. According to the equity theory [15], individuals may become demotivated and decrease their input, or seek change, if they perceive that their efforts are not being justly compensated. The advantages of career development and career planning within an organization include reducing employee turnover rates and increasing job satisfaction among employees [25]. Huyghebaert et al. [26] have examined the relationship between perceived career opportunities and turnover intentions in the nursing context, and their findings suggest that perceived career opportunities can prevent employees from intending to leave their jobs. Therefore, it is reasonable to infer that when younger employees believe that they are not being provided with adequate perceived career opportunities in their current workplace despite their hard work, they may feel a sense of injustice and consequently, become more likely to quit their jobs. In view of the above, the current study proposed the following hypotheses for testing:

**$H_1$**: *PC positively influences JS among young employees.*

**$H_2$**: *PC negatively influences TI among young employees.*

**Perceived pay for performance (PP).**   Gerhart and Fang [27] defined pay for performance as a compensation programme that offers pay according to the performance in terms of outputs (e.g., sales volume) or behavioural evaluation. The correlation between pay growth and performance may not necessarily align with the correlation between employees' perceptions of performance-based pay and actual performance [28], which could affect employee attitudes. Consistent with the equity theory [15], employees' perceptions of equity through the comparison of inputs and outputs affect their attitudes at the workplace. Ren et al. [29] stressed the importance of assessing the influence of PP on employees' attitudes at the workplace and identified PP and pay-level satisfaction as significant determinants of attitudes at the workplace (e.g., JS). Meanwhile, Kollmann et al. [30] described the differences in JS between younger and older employees according to the monetary rewards, task contributions, and imbalances in the relationship between monetary rewards and task contributions. In a more recent study, Bae [31] reported the significant and positive influence of perceived fairness of performance evaluation on pay satisfaction, organisational satisfaction, and JS.

Additionally, the JD-R model has posited that job resources (e.g. satisfying salary, appreciation and performance feedback) have motivational potential and may therefore lead to work engagement, which may result in positive organizational outcomes, including retention intentions [19]. The motivational impact of fair performance appraisal may be attenuated if employees perceive the performance appraisal process as lacking in fairness, validity, and reliability [32,33]. As employees highly value distributive justice on PP, they tend to view their job continuity to be more predictable, controllable, and secure, which reduce the formation of TI [34]. Hazeen and Umarani [35] have provided empirical evidence that employees' perceived fairness of human resource practices, such as performance management and compensation, has a significant positive association with their intention to remain with the organization. In line with the equity theory, TI can be viewed as the resultant outcome of perceived inequity [16]. Considering that, creating a fair working environment serves as a key strategy for the management to reduce TI among the employees. Based on the findings of prior studies, the following hypotheses were proposed for testing:

**$H_3$**: *PP positively influences JS among young employees.*

**$H_4$**: *PP negatively influences TI among young employees.*

**Affective organisational commitment (AC).** Meyer et al. [36] and Buitendach and De Witte [37] described AC as how employees are emotionally attached to, identify with, and involved with the organisation. AC is an individual attitudinal response that develops over time in response to an individual's employment experiences and beliefs about the work environment [38], which in turn influences individuals' work attitudes [39]. Several studies have demonstrated that employees who exhibit affective commitment are intrinsically motivated, passionate about achieving organizational goals [40,41]. Nauman et al. [42] have pointed out the social exchange process by which training enhances the AC of employees which ultimately boosts employee's job satisfaction. Moreover, The impact of AC has been the focus of research in the field of organizational behaviour and has been proved to be intricately linked to both individual behaviour and organizational outcomes [39]. Gara et al. [43] highlighted the significant and negative influence of AC on TI. Employees tend to develop emotional attachments with their organization when they experience positive interpersonal relationships with their colleagues and supervisors and perceive a congruence between their personal values and the organization's values [44,45], thus resulting in a decreased likelihood of turnover intentions among employees [46]. Hence, it is rational to infer that employees who possess AC demonstrate a heightened sense of belonging and connection to the organization's vision, coupled with a strong desire to remain with the organization. Such emotional attachment to the organization may also lead employees to subconsciously perceive their work as meaningful. Thus, the following hypotheses were tested in this study:

$H_5$: *AC positively influences JS among young employees.*

$H_6$: *AC negatively influences TI among young employees.*

**Job satisfaction (JS).** Purani and Sahadev [47] described JS as one's sense of fulfilment gained from policies at the workplace, which are related to human resources, compensation, supervision, task clarity, and career development. Lin et al. [48] have suggested that enhancing JS can yield multiple benefits for organizations, including decreased employee turnover and increased operational efficiencies, leading to cost savings. Bharadwaj et al. [49] have proposed compelling evidence to support the notion that job satisfaction serves as a crucial antecedent to employee retention. This finding is supported by the JD-R model which posits the motivational process that job resources could stimulate positive work outcomes such as retention intention through positive work-related state (e.g. work engagement) or satisfaction of basic needs [19,50]. Additionally, some researchers have attempted to establish a relationship between JS and turnover intention. A potentially effective strategy for organizations to decrease employee turnover intention is to cultivate job satisfaction [51]. Alam and Asim [11] have reported the significant and negative influence of different dimensions of job satisfaction (e.g. satisfaction with supervision, compensation, and career development) on TI. It is rational to infer employees who report higher levels of JS are more inclined to stay in their current workplace. Thus, the current study tested the following hypothesis:

$H_7$: *JS negatively influences TI among young employees.*

**Job autonomy (JA).** Spector [52] described JA as the extent to which employees can make their own decisions based on their judgment and preferences on how they execute the assigned tasks. Specifically, when employees have autonomy at work, they can complete tasks in a free manner according to their judgment and preferences. The job characteristics model [53] proposed that core job characteristics including autonomy generate a sense of responsibility for

outcomes, meaningfulness, and knowledge of results, which in turn elicits intrinsic work motivation and job satisfaction. Autonomous jobs are expected to encourage higher levels of job satisfaction than controlled jobs because JA makes employees feel self-determined and free from external controls or constraints [54]. As a job resource, JA can contribute to positive work outcomes [19]. Consistent with the studies above, Morgeson and Humphrey [55] have pointed out that JA is an important contextual resource to positively influence employees' attitudes and behaviours. Thus, the current study expects a boosting effect of job autonomy on job satisfaction.

Previous studies have mainly examined the moderating role of job resources on the relationship between job demands and work outcomes [56–59]. Hakanen et al. [60] also suggested that resources like job autonomy could moderate the impact of job demands for the majority of jobs. Prior researches have endeavoured to elucidate the degree to which job resources, such as job autonomy, act as a buffer in the association between variables, namely the relationship between job demands and work outcomes. Nevertheless, it is noteworthy that scant studies have explored the moderating effect of job resources in relationships between positive variables. Dominguez et al. [61] have identified a need for additional research on the moderating influence of job resources in the relationships among positive variables. In response to this call, Wan and Duffy [12] have investigated the moderating effect of JA on the association between decent work and job satisfaction, discovering that JA can strengthen the positive link between these two variables. According to Dominguez et al. [61], promoting elements of intrinsic motivation can enhance the relationship between intrinsic motivational resources and positive work attitudes, including work engagement and job satisfaction. Therefore, it is rational to infer that JA, as an intrinsic motivational organizational resource, may boost the effects of other job resources such as PC, PP and AC on job satisfaction. Additionally, this boost effect may stem from the accumulation of job resources, as proposed by Hobfoll [62], wherein workers who have greater autonomy experience a stronger motivational effect from job resources compared to those with lower autonomy levels. In high-autonomy jobs, young individuals who have greater decision-making power and place a premium on career development and performance-based pay are given greater freedom to demonstrate their abilities and receive corresponding incentives. This, in turn, leads to increased job satisfaction. Moreover, employees who have high levels of JA feel a stronger sense of connection to their organization when they are allowed to perform their duties in a way that aligns with their personal values and goals. This sense of connection may ultimately result in higher job satisfaction. Thus, the current study proposed the following hypotheses for testing:

*H$_8$*: *JA positively moderates the relationship between PC and JS.*

*H$_9$*: *JA positively moderates the relationship between PP and JS.*

*H$_{10}$*: *JA positively moderates the relationship between AC and JS.*

Fig 1 presents the conceptual framework of this study.

## Methodology

### Data collection

The young workforce in China generally has lower organisational loyalty and is associated with higher turnover rate [63,64]. Therefore, it was deemed significant for the current study to assess the influence of PC, PP, and AC on JS and TI, as well as the moderating effect of JA among the young employees in China. Referring to the China State Council [65], youths are of those between 14 and 35 years old. Considering the context of the current study, working individuals of between 18 and 35 years old in China were targeted.

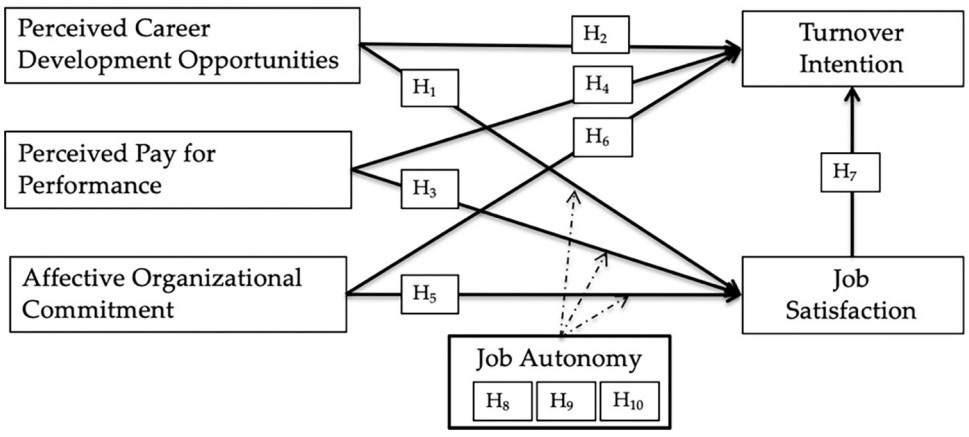

**Fig 1. Conceptual framework.**

This research employed a quantitative approach to examine the relationship between variables. G*Power 3.1 was used to calculate the minimum sample size required to achieve the target analysis level and the desired minimum sample size with a power of 0.95 and an effect size of 0.15, taking into account the five predictors in this study. The output result indicates that at least 138 valid samples are required [66]. However, for partial least squares-structural equation modeling analysis, it is recommended to use a minimum of 200 samples [67]. Online data collection was conducted between July 1, 2022 and August 8, 2022. The online survey was administered through WJX (http://www.wjx.cn/), a functional and user-friendly tool. In order to ensure that all participants were young Chinese adults who were employed and aligned with the background of this study, judgmental questions were used for exclusion purposes. Prior to the commencement of the formal questionnaire, all participants agreed to sign a written informed consent form, in which they were informed of the purpose of the data collection, the final destination of the data, and their right to withdraw from participation at any time. The local ethics committee (Ethical Review of Biomedical Research Involving Human Beings) ruled that no formal ethics approval was required because the data is anonymous, not sensitive or confidential in nature and no vulnerable or dependent groups included. At the end of the data collection, this study successfully obtained 532 valid questionnaire sets.

## Measures

The measurement items were adopted from prior related studies. The English version of these measurement items is presented in the *Supplementary Material 1*: *Survey Instrument*. The final scale was developed in English, which was later translated into Chinese. Two experts were appointed to evaluate the final version of the developed scale to ensure the accuracy and validity of these items and the equivalence of measures in both languages. Respondents were required to provide their responses according to a five-point Likert scale, with the endpoints of "strongly disagree" (1) and "strongly agree" (5).

## Common Method Variance (CMV)

Harman's single factor test was employed as an indicative procedure in this study to examine any potential CMV issues [68]. The single factor accounted for 32.75% ($< 50\%$), which indicated the insignificance of the CMV issues. This study also performed the full collinearity test [69] to evaluate CMV. Accordingly, the obtained values of variance inflation factor (VIF) for

the input variables must be lower than the threshold value of 3.3 to establish the absence of the multicollinearity issue in the data. Based on the obtained results, the VIF values for PC (1.240), PP (1.221), AC (1.301), JS (1.221), JA (1.190), and TI (1.988) did not exceed 3.3, suggesting the absence of collinearity in this study's data.

## Multivariate normality

It is crucial to examine the multivariate normality of the data for the selection of appropriate data analysis method [70]. Using the Web Power tool, this study obtained Mardia's multivariate skewness and kurtosis and p-values for assessment. The recorded p-values did not exceed 0.05, suggesting the issue of multivariate non-normality in this study's data.

## Data analysis

PLS-SEM was performed in this study. PLS-SEM is a causal modelling approach that maximises the explained variance of the dependent latent constructs [71]. Considering the presence of non-normality issues in this study, variance-based PLS-SEM estimation was considered. Prior to the assessment of the structural model, this study examined structural reliability, convergence validity, and discrimination validity. In particular, the reliability of the items was assessed based on Cronbach's alpha, while the internal consistency reliability was determined based on Dijkstra-Hensele's rho and composite reliability. Meanwhile, convergent validity was tested based on the values of average variance extracted (AVE). Besides that, discriminant validity was measured based on the Fornell-Larcker criterion, heterotrait-monotrait ratio (HTMT), and loadings and cross-loadings. Following that, the paths between constructs were estimated.

## Findings

### Demographic profile of respondents

Table 1 presents the demographic profile of respondents in this study. In particular, 49.2% of the total respondents were male respondents, and the remaining 50.8% were female respondents. Most of the respondents were of 26 to 30 years of age (41.4%), followed by those in the age group of between 31 and 35 years (40.2%). Only 18.4% of the total respondents were of between 18 and 25 years old. Besides that, 51.7% of the total respondents reported working at the current workplace for six to 12 years. Other respondents reported working at the current workplace for less than two years (22.2%) and for two to five years (20.5%). Only 5.6% of the total respondents worked at the current workplace for 13 to 17 years. Most of the respondents had less than 10 days of annual paid leave (63.3%). Other respondents revealed to have more than 30 days of annual paid leave (6.6%). As for their monthly income, 43.0% of the total respondents reported earning between CNY 5,000 and CNY 8,000, followed by those who earned between CNY 8,000 and CNY 11,001 (37.2%), between CNY 11,001 and CNY 14,000 (7.7%), less than CNY 5,000 (7.7%), and lastly, more than CNY 14,000 (4.3%).

### Reliability and validity

It is important to first evaluate the reliability and validity of the developed instrument prior to the assessment of the measurement model [67]. Hair et al. [72] recommended the use of Cronbach's alpha, composite reliability, and Dijkstra-Hensele's rho for the evaluation of internal consistency reliability. The results in Table 2 revealed that all values of Cronbach's alpha, composite reliability, and Dijkstra-Hensele's rho exceeded the recommended threshold value of 0.7, confirming the internal consistency reliability of the instrument. Besides that, this study

**Table 1. Demographic profile of respondents.**

| | N | % | | N | % |
|---|---|---|---|---|---|
| *Gender* | | | *Education* | | |
| Male | 262 | 49.2 | Secondary school certificate | 40 | 7.5 |
| Female | 270 | 50.8 | Diploma/technical school certificate | 138 | 25.9 |
| Total | 532 | 100.0 | Bachelor degree or equivalent | 234 | 44.0 |
| | | | Master's degree | 105 | 19.7 |
| *Age* | | | Doctoral degree | 15 | 2.8 |
| 18–25 years | 98 | 18.4 | Total | 532 | 100.0 |
| 26–30 years | 220 | 41.4 | | | |
| 31–35 years | 214 | 40.2 | *Types of Organizations* | | |
| Total | 532 | 100.0 | Information Technology | 109 | 20.5 |
| | | | Finance | 138 | 25.9 |
| Marital Status | | | Construction | 117 | 22.0 |
| Single | 273 | 51.3 | Media | 97 | 18.2 |
| Married | 248 | 46.6 | Education | 55 | 10.3 |
| Divorced | 8 | 1.5 | Others | 16 | 3.0 |
| Widowed | 3 | .6 | Total | 532 | 100.0 |
| Total | 532 | 100.0 | | | |
| | | | *Years of Working* | | |
| *Positions* | | | Less than 2 years | 118 | 22.2 |
| Technical staff | 228 | 42.9 | 2–5 years | 109 | 20.5 |
| Administration staff | 123 | 23.1 | 6–12 years | 275 | 51.7 |
| Production staff | 48 | 9.0 | 13–17 years | 30 | 5.6 |
| Sales staff | 130 | 24.4 | Total | 532 | 100.0 |
| Top management | 3 | .6 | | | |
| Total | 532 | 100.0 | *Average Monthly Income* | | |
| | | | Less than CNY5000 | 41 | 7.7 |
| *Annual Paid Leave* | | | CNY5000-CNY8000 | 229 | 43.0 |
| Less than 10 days | 337 | 63.3 | CNY8001-CNY11000 | 198 | 37.2 |
| 11–20 days | 112 | 21.1 | CNY11001-CNY14000 | 41 | 7.7 |
| 21–30 days | 48 | 9.0 | More than CNY14000 | 23 | 4.3 |
| More than 30 days | 35 | 6.6 | Total | 532 | 100.0 |
| Total | 532 | 100.0 | | | |

**Table 2. Reliability and validity.**

| Variables | No. Items | Mean | Standard Deviation | Cronbach's Alpha | Dijkstra-Hensele's *rho* | Composite Reliability | Average Variance Extracted | Variance Inflation Factors |
|---|---|---|---|---|---|---|---|---|
| PC | 5 | 3.261 | 1.154 | 0.931 | 0.939 | 0.948 | 0.786 | 1.148 |
| PP | 5 | 3.319 | 1.164 | 0.929 | 0.939 | 0.947 | 0.782 | 1.130 |
| AC | 5 | 3.236 | 1.188 | 0.940 | 0.946 | 0.955 | 0.809 | 1.131 |
| JS | 5 | 3.268 | 1.184 | 0.935 | 0.944 | 0.951 | 0.794 | 1.100 |
| JA | 5 | 3.325 | 1.161 | 0.931 | 0.940 | 0.948 | 0.786 | 1.101 |
| TI | 6 | 2.541 | 1.068 | 0.935 | 0.941 | 0.949 | 0.758 | - |

**Note.** PC: Perceived Career Development Opportunities; PP: Perceived Pay for Performance; AC: Affective Organizational Commitment; JS: Job Satisfaction; JA: Job Autonomy; TI: Turnover Intention

**Table 3. Discriminant validity.**

|  | PC | PP | AC | JS | JA | TI |
|---|---|---|---|---|---|---|
| *Heterotrait-monotrait ratio (HTMT)* |  |  |  |  |  |  |
| PC | - |  |  |  |  |  |
| PP | 0.195 | - |  |  |  |  |
| AC | 0.291 | 0.216 | - |  |  |  |
| JS | 0.190 | 0.245 | 0.218 | - |  |  |
| JA | 0.224 | 0.257 | 0.167 | 0.205 | - |  |
| TI | 0.461 | 0.442 | 0.505 | 0.450 | 0.412 | - |
| *Fornell-Larcker criterion* |  |  |  |  |  |  |
| PC | 0.887 |  |  |  |  |  |
| PP | 0.184 | 0.884 |  |  |  |  |
| AC | 0.273 | 0.205 | 0.899 |  |  |  |
| JS | 0.179 | 0.233 | 0.206 | 0.891 |  |  |
| JA | 0.208 | 0.239 | 0.158 | 0.194 | 0.887 |  |
| TI | -0.432 | -0.414 | -0.476 | -0.422 | -0.383 | 0.871 |

**Note.** PC: Perceived Career Development Opportunities; PP: Perceived Pay for Performance; AC: Affective Organizational Commitment; JS: Job Satisfaction; JA: Job Autonomy; TI: Turnover Intention.

measured both convergent validity and discriminant validity. In particular, convergent validity implies that a set of indicators reflects the same construct, which can be determined based on the values of average variance extraction [67]. The obtained results revealed that all values of AVE exceeded the threshold value of 0.5, suggesting adequate convergent validity. The results further revealed that all VIF values of constructs did not exceed 3.3, confirming the absence of multicollinearity issues [69].

As for the measurement of discriminant validity, this study referred to the HTMT ratio, Fornell-Larcker criterion, and cross-loadings. Referring to Table 3, all HTMT values did not exceed 0.9, suggesting the discriminant validity of the study's data. The recorded values of Fornell-Larcker criterion revealed that the square root value of AVE for each latent variable (the diagonal values) exceeded the square root of other variables [73]. The results demonstrated adequate discriminant validity.

Meanwhile, the results in Table 4 revealed loadings of higher than 0.5, which exceeded the cross-loadings. These results reaffirmed the adequacy of discriminant validity of all items [67].

## Testing of hypotheses

The structural model was assessed based on the coefficient of determination ($r^2$) and effect size ($f^2$). Accordingly, the values of $r^2$ range from 0 to 1, with a higher value indicating higher explanatory power. Referring to Table 5, JS recorded $r^2$ of 0.105, indicating that PC, PP, and AC explained 10.5% of the total variation in JS. On the other hand, TI recorded $r^2$ of 0.466, indicating that PC, PP, AC, and JS explained 46.6% of the total variation in TI. Meanwhile, PC ($f^2$ of 0.113), PP ($f^2$ of 0.102), JS ($f^2$ of 0.112), and AC ($f^2$ of 0.151) exhibited small to medium effects on TI.

Referring to the results tabulated in Table 5 and Fig 2, this study demonstrated the significant and positive influence of PC ($H_1$: $\beta = 0.085$, $p < 0.05$), PP ($H_3$: $\beta = 0.161$, $p < 0.05$), and AC ($H_5$: $\beta = 0.128$, $p < 0.05$) on JS. This study also demonstrated the significant and negative influence of PC ($H_2$: $\beta = -0.259$, $p < 0.05$), PP ($H_4$: $\beta = -0.245$, $p < 0.05$), AC ($H_6$: $\beta = -303$, $p < 0.05$), and JS ($H_7$: $\beta = -0.257$, $p < 0.05$) on TI. This study found no zero between the 5% CI and 95% CI for all proposed relationships, confirming the support of these hypotheses.

**Table 4. Loadings and cross-loadings.**

|  | PC | PP | AC | JS | JA | TI |
|---|---|---|---|---|---|---|
| PC1 | 0.983 | 0.191 | 0.264 | 0.176 | 0.205 | -0.436 |
| PC2 | 0.860 | 0.195 | 0.265 | 0.179 | 0.210 | -0.385 |
| PC3 | 0.857 | 0.150 | 0.236 | 0.115 | 0.170 | -0.350 |
| PC4 | 0.861 | 0.129 | 0.225 | 0.156 | 0.154 | -0.345 |
| PC5 | 0.865 | 0.143 | 0.217 | 0.161 | 0.177 | -0.390 |
| PP1 | 0.183 | 0.979 | 0.216 | 0.236 | 0.239 | -0.423 |
| PP2 | 0.168 | 0.862 | 0.180 | 0.216 | 0.198 | -0.353 |
| PP3 | 0.136 | 0.868 | 0.148 | 0.187 | 0.214 | -0.346 |
| PP4 | 0.144 | 0.838 | 0.158 | 0.132 | 0.189 | -0.332 |
| PP5 | 0.176 | 0.867 | 0.193 | 0.243 | 0.213 | -0.367 |
| AC1 | 0.272 | 0.218 | 0.984 | 0.206 | 0.169 | -0.478 |
| AC2 | 0.240 | 0.167 | 0.871 | 0.202 | 0.132 | -0.403 |
| AC3 | 0.270 | 0.195 | 0.881 | 0.134 | 0.116 | -0.414 |
| AC4 | 0.240 | 0.181 | 0.879 | 0.198 | 0.134 | -0.446 |
| AC5 | 0.202 | 0.155 | 0.876 | 0.182 | 0.154 | -0.392 |
| JS1 | 0.173 | 0.243 | 0.208 | 0.983 | 0.195 | -0.438 |
| JS2 | 0.168 | 0.220 | 0.197 | 0.881 | 0.195 | -0.388 |
| JS3 | 0.164 | 0.177 | 0.189 | 0.868 | 0.169 | -0.368 |
| JS4 | 0.127 | 0.178 | 0.163 | 0.852 | 0.115 | -0.336 |
| JS5 | 0.163 | 0.214 | 0.155 | 0.866 | 0.183 | -0.340 |
| JA1 | 0.193 | 0.228 | 0.156 | 0.193 | 0.977 | -0.380 |
| JA2 | 0.198 | 0.200 | 0.111 | 0.157 | 0.866 | -0.323 |
| JA3 | 0.186 | 0.194 | 0.120 | 0.180 | 0.871 | -0.316 |
| JA4 | 0.159 | 0.220 | 0.170 | 0.181 | 0.866 | -0.326 |
| JA5 | 0.190 | 0.217 | 0.139 | 0.144 | 0.847 | -0.356 |
| TI1 | -0.431 | -0.406 | -0.495 | -0.415 | -0.379 | 0.978 |
| TI2 | -0.347 | -0.321 | -0.371 | -0.361 | -0.343 | 0.851 |
| TI3 | -0.367 | -0.349 | -0.412 | -0.382 | -0.317 | 0.866 |
| TI4 | -0.394 | -0.380 | -0.426 | -0.339 | -0.305 | 0.862 |
| TI5 | -0.348 | -0.329 | -0.375 | -0.377 | -0.359 | 0.835 |
| TI6 | -0.365 | -0.372 | -0.395 | -0.331 | -0.298 | 0.823 |

**Note.** PC: Perceived Career Development Opportunities; PP: Perceived Pay for Performance; AC: Affective Organizational Commitment; JS: Job Satisfaction; JA: Job Autonomy; TI: Turnover Intention

However, the moderating effect of JA was found statistically insignificant at 0.05 level of significance, indicating that JA did not moderate the relationship between PC and JS in this study. Likewise, the results indicated that JA did not moderate the influence of PP and AC on JS. Thus, $H_8$, $H_9$, and $H_{10}$ were rejected.

## Importance-performance map analysis

When it comes to PLS-SEM, the importance-performance map analysis (IPMA) presents valuable insights on the influence of antecedent constructs and their relevance for managerial actions [74]. Through this map analysis, significant input constructs, which basically exhibit strong overall effects but with low average latent variable scores, can be identified. The most effective managerial actions can be determined by focusing on low-performing but important constructs. Referring to Fig 3, the results of the overall effects (importance) and average latent

**Table 5. Hypotheses testing.**

| Hypo | | Beta | CI Min | CI Max | t Value | p value | $r^2$ | $f^2$ | Decision |
|---|---|---|---|---|---|---|---|---|---|
| H₁ | PC → JS | 0.085 | 0.012 | 0.158 | 1.931 | 0.027 | | 0.007 | Supported |
| H₃ | PP → JS | 0.161 | 0.082 | 0.236 | 3.416 | 0.000 | 0.105 | 0.026 | Supported |
| H₅ | AC → JS | 0.128 | 0.051 | 0.205 | 2.752 | 0.003 | | 0.016 | Supported |
| H₂ | PC → TI | -0.259 | -0.321 | -0.198 | 6.919 | 0.000 | | 0.113 | Supported |
| H₄ | PP → TI | -0.245 | -0.312 | -0.181 | 6.173 | 0.000 | 0.466 | 0.102 | Supported |
| H₆ | AC → TI | -0.303 | -0.367 | -0.237 | 7.619 | 0.000 | | 0.151 | Supported |
| H₇ | JS → TI | -0.257 | -0.317 | -0.197 | 7.052 | 0.000 | | 0.112 | Supported |
| | JA → JS | 0.116 | 0.051 | 0.189 | 2.767 | 0.003 | | | |
| H₈ | JA*PC → JS | -0.008 | -0.071 | 0.059 | 0.194 | 0.423 | | No Moderation | |
| H₉ | JA*PP → JS | -0.040 | -0.113 | 0.032 | 0.893 | 0.186 | | No Moderation | |
| H₁₀ | JA*AC → JS | 0.019 | -0.051 | 0.088 | 0.445 | 0.328 | | No Moderation | |

**Note.** PC: Perceived Career Development Opportunities; PP: Perceived Pay for Performance; AC: Affective Organizational Commitment; JS: Job Satisfaction; JA: Job Autonomy; TI: Turnover Intention.

variable scores (performance) clearly revealed AC as the most important construct in relation to TI, followed by PP, PC, JS, and lastly, JA. Despite the insignificant difference in the performance of all these constructs, AC recorded the lowest performance. In other words, the performance of the aspects related to AC should be prioritised due to its highest importance and lowest performance.

## Discussion and implications

This cross-sectional study exclusively focused on turnover intention of young employees in China. Retaining young employees is undoubtedly a major challenge for organisations today. Therefore, it is crucial for organisations to understand how attached their employees are to the workplace. Focusing on that, the current study presented empirical evidence on factors that influence TI. Firstly, this study empirically proved the direct significant influence of perceived career development opportunities on job satisfaction (H₁) and turnover intention (H₂). Stahl et al. [75] and Ohunakin et al. [76] presented similar findings on the substantial influence of perceived career development opportunities on turnover intention. Likewise, Price and Reichert [22] identified perceived career development opportunities as one of the factors that can enhance job satisfaction. In other words, PC significantly influences JS and TI. This study also empirically demonstrated the significant and positive influence of perceived pay for performance on job satisfaction (H₃). This finding was consistent with the studies conducted by Ren et al. [29] and Bae [31], which have reported the significant and positive relationship between perceived pay for performance and job satisfaction. This finding highlighted the importance of perceived pay for performance in motivating young Chinese employees and improving job satisfaction. Additionally, the study discovered that perceived pay for performance had a negative impact on turnover intention, providing support for H₄. This finding aligned with the assertions by Kuvaas et al. [77] and Hur and Ha [34]. They pointed out performance-based pay had a negative impact on turnover intention. In other words, among young Chinese employees, the recognition and appropriate compensation of their efforts and contributions may lead to a decreased likelihood of contemplating job departure.

Besides that, this study empirically demonstrated the positive influence of affective organizational commitment on job satisfaction. Thus, H₅ was supported. This finding was consistent

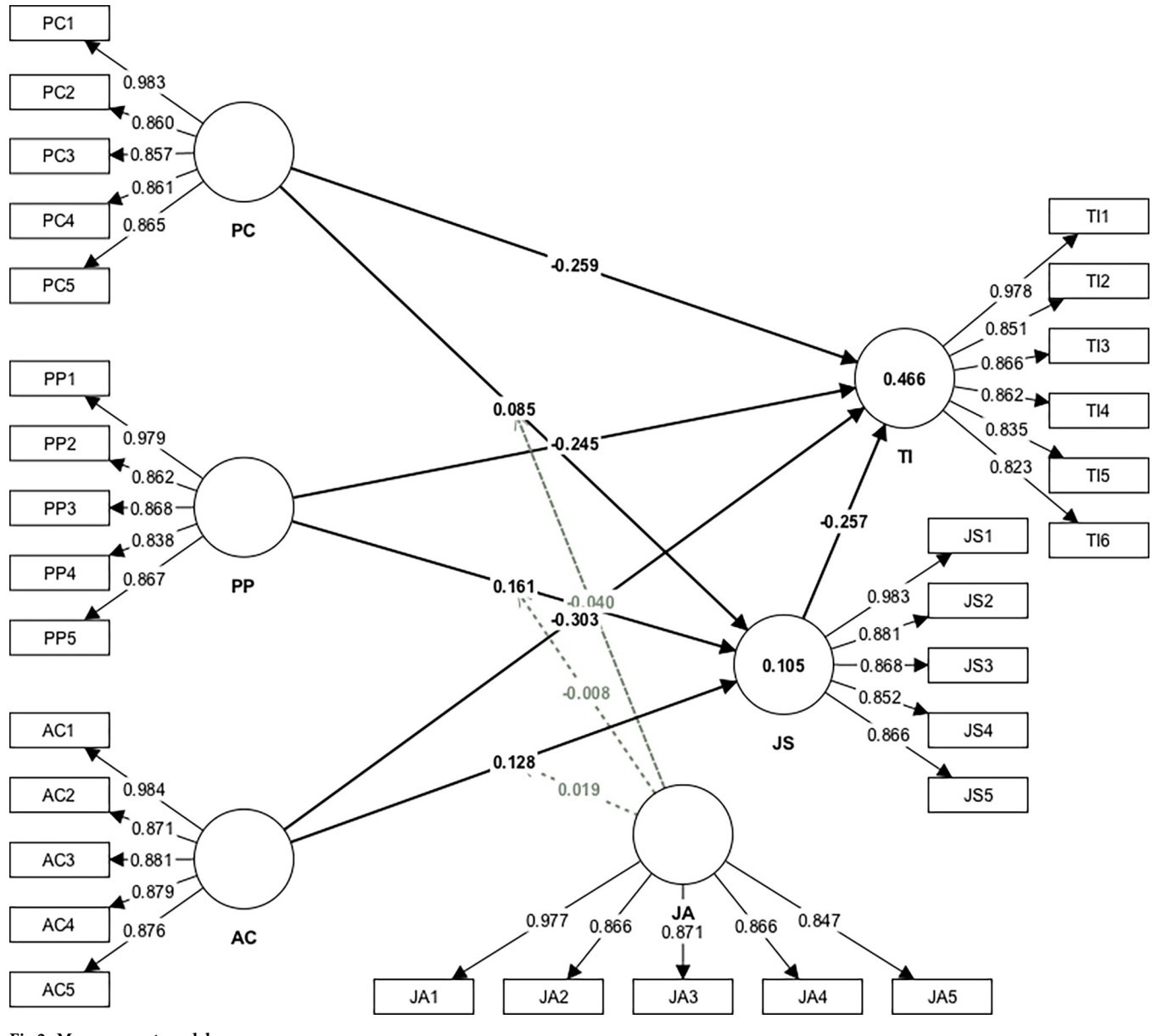

**Fig 2. Measurement model.**

with several prior studies [39,78,79]. In other words, employees who have a strong emotional attachment to their organization are more likely to experience job satisfaction. Furthermore, the current study shed light on the relationship between affective organizational commitment and turnover intention in the context of young Chinese workers. Specifically, the study revealed that employees' affective organizational commitment negatively influenced their turnover intentions (H$_6$). Gara et al. [43] and Ampofo and Karatepe [44] presented similar findings on the substantial influence of affective organizational commitment on turnover intention. The current study adds to the existing literature on the importance of job resources such as affective organizational commitment in reducing employee turnover intention, which has implications for organizations seeking to improve their retention rate. The obtained results of this study also confirmed job satisfaction as a significant determinant of turnover intention

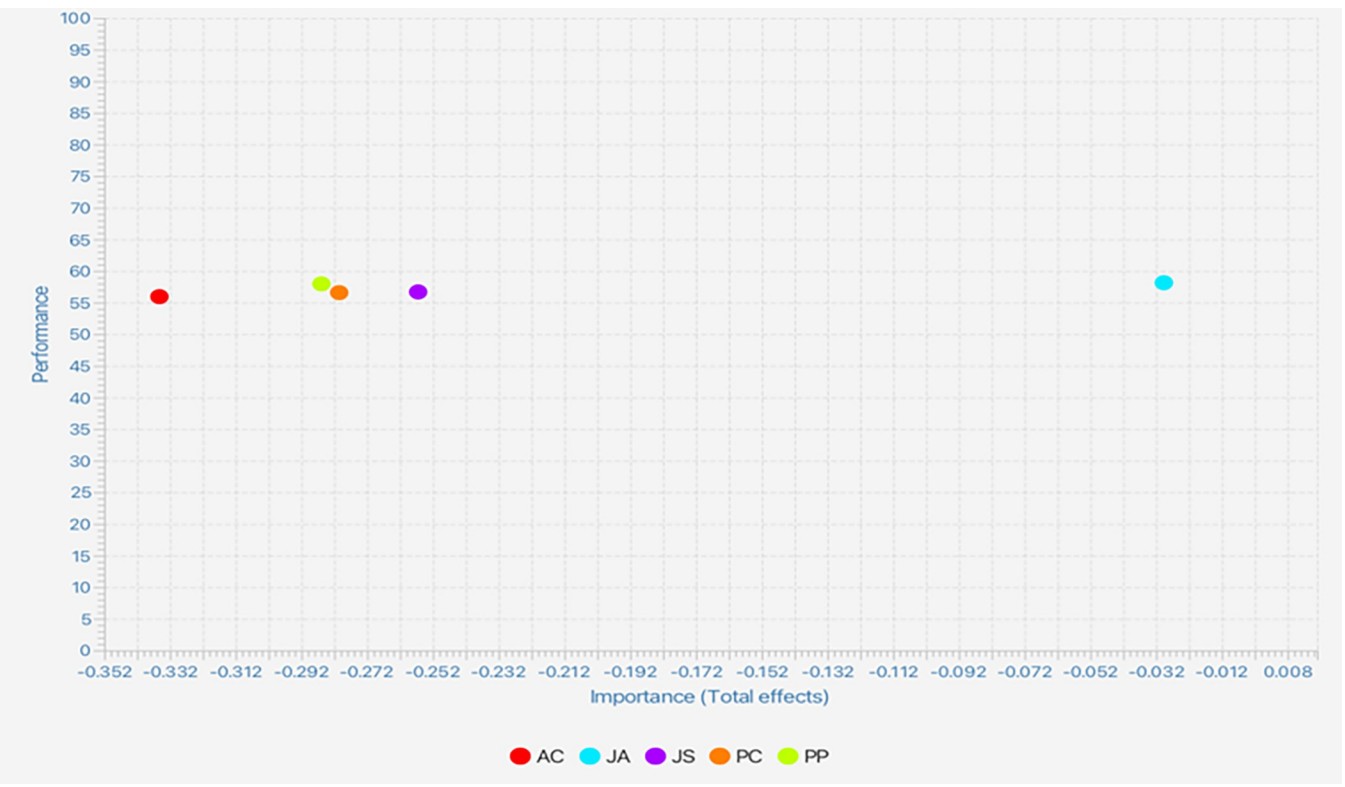

**Fig 3. Importance performance map.**

(H$_7$), which supported the findings of several prior studies [11,12]. In other words, the significance of job satisfaction as a predictor of turnover intention should not be overlooked.

Dominguez et al. [61] have highlighted the necessity for additional research on the moderating impact of job resources in the relationships between positive variables. While previous studies have primarily concentrated on the buffering effects of job resources on job demands, little attention has been given to the moderating role of job resources. Consequently, there is an urgent need for more extensive research to gain a more comprehensive understanding of this complex interplay. Responding to this call, this study empirically proved the insignificant moderating effect of job autonomy on the relationships of perceived career development opportunities (H$_8$), perceived pay for performance (H$_9$), and affective organizational commitment (H$_{10}$) with job satisfaction. Tang and Vandenberghe [80] found that affective organizational commitment may make employees feel driven by task autonomy, which can lead to job burnout, dissatisfaction, and intention to leave, and that employee job autonomy did not have a significant effect in this process; this is the same as the results of this study, where employee job autonomy did not moderate the effect between affective organizational commitment and employee job satisfaction. Wang et al. [2] confirmed that employees' job satisfaction moderates the relationship between employees' job autonomy and turnover intention, and found a significant relationship between employees' salary factors (such as performance pay) and job satisfaction and it was not moderated by employees' job autonomy; this is consistent with the results of H$_9$.

In other words, young employees take into account the importance of job autonomy at the workplace, but different levels of job autonomy do not affect the influence of these factors on job satisfaction. This finding contrasted with previous studies, such as Iplik [81], which

showed that job autonomy moderates the relationship between employees' perceptions and job satisfaction. Similarly, Wan and Duffy [12] found that job autonomy moderated the relationship between job resources, such as decent work, and job satisfaction. It is possible that different demographic groups could yield different results, and cultural or contextual differences may explain the current findings. Cultural differences can influence how individuals perceive and respond to job resources, such as job autonomy, which could affect the moderating role of job autonomy in the relationship between other job resources and job satisfaction. In contrast to countries that prioritize individualism, Chinese culture emphasizes group harmony and collectivism, which could lead to greater acceptance of limited job autonomy and a focus on fulfilling assigned responsibilities [82]. Therefore, future research is encouraged to explore the moderating effect of job autonomy between positive variables in diverse contexts.

### Theoretical implications

This paper supports the call for research into the impact of employees' interpretations of HRM practices on the effectiveness of the HRM system [29,83]. This study presented significant theoretical implications that would be of much interest for future research on turnover intention, particularly among young employees. This study addressed the call of prior studies [84] on the need for more in-depth insights on how job characteristics influence the formation of work attitudes among employees. This study presented empirical evidence on factors that can reduce young employees' turnover intention in relation to the unique attributes of this young workforce. In particular, the significance of perceived career development opportunities, perceived pay for performance and affective organizational commitment in influencing turnover intention was highlighted. This study demonstrated the importance of considering the work characteristics of young employees when it comes to exploring the formation of their job satisfaction and turnover intention. This study also presented unexpected findings on how job autonomy does not moderate the relationships of perceived career development opportunities, perceived pay for performance and affective organizational commitment with job satisfaction despite the importance of job autonomy for young employees at the workplace.

### Practical implications

The results of this study hold significant practical implications for organizations, particularly those operating in China, seeking to reduce turnover intention among young employees. To attain this objective, organizations must concentrate on augmenting perceived career development opportunities, perceived pay for performance, and affective organizational commitment. To improve employee perceptions of career development opportunities, organizations can facilitate training and development programs designed to enhance employees' skills and competencies, provide opportunities for job rotation, and mentorship programs. Furthermore, managers can implement performance-based pay systems, furnish regular feedback and recognition to employees, and cultivate a culture of accountability and transparency. Such measures will elevate employees' perception of fairness in the pay structure and stimulate their motivation to perform well.

This study also demonstrated affective organizational commitment as the most important element with the highest overall effects on turnover intention among young employees in China. However, it is the least-valued element by the management. Therefore, organizations should prioritize the development of strategies and initiatives that enhance employees' affective commitment to the organization. Organizations can create a positive work environment that fosters trust, openness, and communication [85]. This can be achieved by promoting a

strong organizational culture, providing opportunities for employee involvement and participation in decision-making, and recognizing and rewarding employees' contributions.

## Conclusion

In summary, this study's results demonstrate the importance of perceived career development opportunities, perceived pay for performance, and affective organizational commitment in determining employee job satisfaction and turnover intention among young Chinese workers. The positive impact of these factors on job satisfaction suggests that employees who perceive opportunities for career advancement and fair rewards for their performance are more likely to experience job satisfaction. Conversely, the negative effect on turnover intention highlights the critical role these factors play in young employee retention within the organization. The current study also discovers that while job autonomy may not directly moderate the relationships between job resources and job satisfaction, it is still an important factor to consider as it can impact employee motivation and engagement [54]. Hence, this study provides important insights for organizations looking to improve employee retention and job satisfaction, and highlights the need for continued research in this area to further explore the complex relationships between these variables.

## Limitations and recommendations

This study encountered several limitations. Firstly, the quantitative approach was adopted in this study, which provided limited understanding on this particular phenomenon. Therefore, it is recommended for future research to consider adopting mixed-methods approach to obtain more comprehensive understanding on this phenomenon. Secondly, as this study used a cross-sectional design, causal relationships among the study variables cannot be ascertained. Therefore, future studies should use longitudinal methods for data collection. Lastly, this study focused on examining the influence of PC, PP, and AC on JS and TI and did not explore the influence of other personal and situational factors. Therefore, it is recommended for future research to incorporate additional constructs into the study's proposed model for testing or implement the same model in other countries for more generalised understanding.

## Supporting information

**S1 Data.**
(CSV)

**S1 File.**
(DOCX)

## Author Contributions

**Conceptualization:** Xuelin Chen, Wan Mohd Hirwani Wan Hussain, Gao Jingzu.

**Formal analysis:** Abdullah Al Mamun, Qing Yang.

**Methodology:** Xuelin Chen, Wan Mohd Hirwani Wan Hussain, Gao Jingzu, Sayed Samer Ali Al Shami.

**Writing – original draft:** Xuelin Chen, Wan Mohd Hirwani Wan Hussain, Sayed Samer Ali Al Shami.

**Writing – review & editing:** Abdullah Al Mamun, Gao Jingzu, Qing Yang.

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
