## [Decision Letter · Decision Letter 0]

28 Feb 2023

PONE-D-23-03002Career Development Opportunity, Remuneration, and Organisational Commitment on Job Satisfaction and Turnover Intention: Moderating Effect of Job AutonomyPLOS ONE

Dear Dr. Al Mamun,

Thank you for submitting your manuscript to PLOS ONE. After careful consideration, we feel that it has merit but does not fully meet PLOS ONE’s publication criteria as it currently stands. Therefore, we invite you to submit a revised version of the manuscript that addresses the points raised during the review process.

We look forward to receiving your revised manuscript.

Kind regards,

Bo Pu, Ph.D.

Academic Editor

PLOS ONE

2. Please include the information provided on ethical considerations and participant consent in the methods section of the manuscript.

3. Please ensure that you include a title page within your main document. You should list all authors and all affiliations as per our author instructions and clearly indicate the corresponding author.

Additional Editor Comments:

Dear Abdullah Al Mamun,

Thank you for submitting your manuscript, Manuscript Number PONE-D-23-03002 and entitled "Career Development Opportunity, Remuneration, and Organisational Commitment on Job Satisfaction and Turnover Intention: Moderating Effect of Job Autonomy," to the PLOS ONE.

I find the topic of career development and organization behavior to be interesting and relevant, I have decided to major revision. Looking forward your modified version.

Best Wishes,

Bo Pu, Ph.D.

Academic Editor

PLOS ONE

Reviewers' comments:

Reviewer's Responses to Questions

**Comments to the Author**

1. Is the manuscript technically sound, and do the data support the conclusions?

Reviewer #1: Yes

Reviewer #2: Yes

2. Has the statistical analysis been performed appropriately and rigorously? 

Reviewer #1: Yes

Reviewer #2: Yes

3. Have the authors made all data underlying the findings in their manuscript fully available?

Reviewer #1: Yes

Reviewer #2: No

4. Is the manuscript presented in an intelligible fashion and written in standard English?

Reviewer #1: Yes

Reviewer #2: Yes

5. Review Comments to the Author

Reviewer #1: 1. The introduction part is too long-winded, the author needs to be more targeted to explain the research topic, to comb out the explanation of the research gap

2. The theoretical basis of the moderating effect hypothesis is not well presented. The advancement of hypothesis needs to be based on theoretical basis. The hypothesis of the moderate effect needs to be further deepened. Therefore, I propose to rearrange hypothesis 8, 9, and 10

3. H8, H9, and H10 were rejected. The author needs to explain in detail in the conclusion of the manuscript why the author chose this moderate variable and whether the moderating effect is not valid can be supported by other literature. Otherwise, the selection of this variable would be meaningless.

Reviewer #2: Dear Author

Thank you for sending your paper "Career Development Opportunity, Remuneration, and Organisational Commitment on Job Satisfaction and Turnover Intention: Moderating Effect of Job Autonomy" to this journal "PLOS One". I read your paper and gave my concern down here:

1. I suppose that the topic is too long and can be shorten.

2. The research problem can be made crystal clear. Moreover, the very first sentence of the introduction section started with "it"! What does it mean? 'It' should be clear.

3. The hypothesis on moderating effect needs further improvement.

4. The methodology section seems alright.

5. I missed the gravity of the discussion section in this manuscript. Would you add discussion section here?

Wish you all the best.

6. PLOS authors have the option to publish the peer review history of their article (what does this mean?). If published, this will include your full peer review and any attached files.

Reviewer #1: No

Reviewer #2: **Yes: **Md. Aftab Uddin

---

## [Author Response · Author response to Decision Letter 0]

20 Mar 2023

PONE-D-23-03002

Career Development Opportunity, Remuneration, and Organisational Commitment on Job Satisfaction and Turnover Intention: Moderating Effect of Job Autonomy

PLOS ONE

Dear Dr. Al Mamun,

Thank you for submitting your manuscript to PLOS ONE. After careful consideration, we feel that it has merit but does not fully meet PLOS ONE’s publication criteria as it currently stands. Therefore, we invite you to submit a revised version of the manuscript that addresses the points raised during the review process.

We look forward to receiving your revised manuscript.

Kind regards,

Bo Pu, Ph.D.

Academic Editor

PLOS ONE

1. Please ensure that your manuscript meets PLOS ONE's style requirements, including 

those for file naming. The PLOS ONE style templates can be found at

Author(s) Reply: Thank you for your constructive comments and suggestions. Amended accordingly

2. Please include the information provided on ethical considerations and participant consent in the methods section of the manuscript.

Author(s) Reply: Thank you for your constructive comments and suggestions. Amended accordingly:

Prior to the commencement of the formal questionnaire, all participants signed an informed consent form, in which they were informed of the purpose of the data collection, the final destination of the data, and their right to withdraw from participation at any time. The local ethics committee (Ethical Review of Biomedical Research Involving Human Beings, China) ruled that no formal ethics approval was required because the data is anonymous, not sensitive or confidential in nature and no vulnerable or dependent groups included. At the end of the data collection, this study successfully obtained 532 valid questionnaire sets.

3. Please ensure that you include a title page within your main document. You should list all authors and all affiliations as per our author instructions and clearly indicate the corresponding author.

Author(s) Reply: Thank you for your constructive comments and suggestions. Amended accordingly:

Chen Xuelin1,2, Abdullah Al Mamun*2, Wan Mohd Hirwani Wan Hussain2, Gao Jingzu2, Qing Yang2, Sayed Samer Ali Al Shami3

1School of Business, Jishou University, 416000 Jishou City, Hunan, China. 

2UKM - Graduate School of Business, Universiti Kebangsaan Malaysia, 43600, UKM Bangi, Selangor Darul Ehsan, Malaysia

3Institute of Technology Management and Entrepreneurship, Universiti Teknikal Malaysia Melaka, Melaka, Malaysia

Address: UKM - Graduate School of Business, Universiti Kebangsaan Malaysia, 43600, UKM Bangi, Selangor Darul Ehsan, Malaysia. Email: almamun@ukm.edu.my/mamun7793@gmail.com

Author(s) Reply: Thank you for your constructive comments and suggestions. Amended accordingly:

Prior to the commencement of the formal questionnaire, all participants signed an informed consent form, in which they were informed of the purpose of the data collection, the final destination of the data, and their right to withdraw from participation at any time. The local ethics committee (Ethical Review of Biomedical Research Involving Human Beings, China. Link: http://www.gd.gov.cn/zwgk/wjk/zcfgk/content/post_2530813.html) ruled that no formal ethics approval was required because the data is anonymous, not sensitive or confidential in nature and no vulnerable or dependent groups included. At the end of the data collection, this study successfully obtained 532 valid questionnaire sets.

ADDITIONAL EDITOR COMMENTS:

Dear Abdullah Al Mamun,

Thank you for submitting your manuscript, Manuscript Number PONE-D-23-03002 and entitled "Career Development Opportunity, Remuneration, and Organisational Commitment on Job Satisfaction and Turnover Intention: Moderating Effect of Job Autonomy," to the PLOS ONE.

I find the topic of career development and organization behavior to be interesting and relevant, I have decided to major revision. Looking forward your modified version.

Best Wishes,

Bo Pu, Ph.D.

Academic Editor

PLOS ONE

Reviewers' comments:

Reviewer's Responses to Questions

Comments to the Author

1. Is the manuscript technically sound, and do the data support the conclusions?

Reviewer #1: Yes

Reviewer #2: Yes

2. Has the statistical analysis been performed appropriately and rigorously? 

Reviewer #1: Yes

Reviewer #2: Yes

3. Have the authors made all data underlying the findings in their manuscript fully available?

Reviewer #1: Yes

Reviewer #2: No

4. Is the manuscript presented in an intelligible fashion and written in standard English?

Reviewer #1: Yes

Reviewer #2: Yes

5. Review Comments to the Author

REVIEWER #1: 

Comment 1: 1. The introduction part is too long-winded, the author needs to be more targeted to explain the research topic, to comb out the explanation of the research gap

Author(s) Reply: Thank you for your constructive comments and suggestions, Prof. Amendment highlighted (added – BLUE; removed RED) below:

INTRODUCTION

It has become increasingly crucial for companies to enhance their work efficiency due to the intense market competition. The workforce has substantial influence on the organisational or business success, which explains the significant need for companies to optimise their workforce, particularly the younger workforce considering their high turnover rate and independent nature (Ramlah et al., 2021). With the constant development of the economy and the increasing competition in the labour market, an increasing number of young employees are contemplating job-hopping or leaving their current company. The higher the turnover intention, the higher the likelihood of quitting the job (Jacobs & Roodt, 2008). Belete (2018) described turnover intention as the intention to leave the workplace. The management must have good understanding of the factors that contribute to turnover intention considering how a higher turnover rate would affect the organisational morale, reduce the sense of identity within the company (Joe et al., 2018), and yield negative financial impact since the companies have to allocate resources for recruitment and training and experience brain drain and lower productivity (Belete, 2018).

The older workforce possesses more traditional values, such as contributing to the country and supporting the family, but the younger workforce emphasises more worldly values and embraces self-development as the key purpose of working (Zhao, 2018). In general, the younger workforce demonstrates unique characteristics at work. They have different expectations at work and expect higher expectations on perceived career development opportunity (PC) and perceived pay for performance (PP) than the older workforce (Magni & Manzoni, 2020). Besides that, prior studies demonstrated lower affective organisational commitment (AC) among young employees than older employees (Singh & Gupta, 2015; Salminen & Miettinen, 2019). Adding to that, the younger workforce highly values job autonomy (JA) at the workplace. Morgeson and Humphrey (2006) defined JA as the extent of freedom and independence allowed for employees to arrange their work schedule, make decisions, and select the method of performing tasks at the workplace. The younger workforce is more independent in nature (Ramlah et al., 2021) and may be less loyal towards the company (Singh & Gupta, 2015). They are not willing to obey orders and instructions, as they prefer to work freely and independently (Lee et al., 2017). Moreover, the younger workforce emphasises how they feel, which makes them highly value job satisfaction (JS) at the workplace (Yi et al., 2010; Zhao, 2018). In short, unlike the older workforce, the younger workforce has higher expectations on PC and PP but lower expectations on AC compared with older workforce and display higher concern for JA and JS. 

Prior studies examined the influence of these unique attributes of young workforce on turnover intention (TI) (Lee et al., 2017; Belete, 2018; Alam & Asim, 2019; Wan & Duffy, 2022). However, none of these studies examined these relationships across different generations. As a result, the significance of these attributes as the main predictors of TI among the young workforce has remained unclear. Prior studies mainly focused on the direct influence of PC, PP, and AC on TI, respectively. Only a few studies justified PC, PP, and AC as predictors of TI. Besides that, the relationships of JA and JS with TI have remained underexplored, which were addressed in the current study. These young employees are the future workforce and soon replace the current management. Retaining these young employees can minimise the costs of recruiting and training new talents. Additionally, prior studies have mainly focused on the direct influence of PC, PP, and AC on TI and largely neglected the mechanism of how PC, PP and AC affect TI. Drawing upon the equity theory (Adams, 1965), which proposes that employees' satisfaction levels are contingent upon their perceptions of fairness regarding the relationship between their inputs and outputs, previous study has pointed out the mediating role of job satisfaction (JS) between the relationship between employee perceptions of organizational justice and employee outcomes (Mashi, 2018). Hence, the present study seeks to investigate the mechanism of how PC, PP and AC affect TI by focusing on JS.

Furthermore, limited research has explored whether there are certain conditions under which the effects of PC, PP and AC on job satisfaction and turnover intention most likely emerges. Examining these unexplored boundary conditions is critical, as it may offer valuable insights that the effects of PC, PP, and AC should not be taken for granted. According to Fried and Ferris (1987), when employees perceive more autonomous opportunities provided by the organization, they will have internal positive feelings toward their work. Job autonomy (JA) might be an important contextual resource to influence employees’ attitudes and behaviours (Morgeson & Humphrey, 2006). Hence, this study explores the role of JA within the organization as a boundary condition to the relationships of PC, PP and AC with job satisfaction and turnover intention. Overall, to address these gaps in the literature, the current study seeks to investigate whether PC, PP, and AC indirectly affect turnover intention by influencing job satisfaction under different levels of job autonomy.

Focusing on the young workforce in China, the current study quantitatively assessed the influence of PC, PP, and AC on JS and TI, as well as the moderating effect of JA. In particular, this study addressed the following research questions: (a) Does PC influence TI directly or indirectly under different levels of JA? (b) Does PP influence TI directly or indirectly under different levels of JA? And (c) Does AC influence TI directly or indirectly under different levels of JA? 

Comment 2: The theoretical basis of the moderating effect hypothesis is not well presented. The advancement of hypothesis needs to be based on theoretical basis. The hypothesis of the moderate effect needs to be further deepened. Therefore, I propose to rearrange hypothesis 8, 9, and 10

Author(s) Reply: Thank you for your constructive comments and suggestions, Prof. Amendment highlighted (added – BLUE; removed RED) below:

Job Autonomy

Spector (1986) described JA as the extent to which employees can make their own decisions based on their judgment and preferences on how they execute the assigned tasks. Specifically, when employees have autonomy at work, they can complete tasks in a free manner according to their judgment and preferences. The job characteristics model (Hackman & Oldham, 1975) proposed that core job characteristics including autonomy generate a sense of responsibility for outcomes, meaningfulness, and knowledge of results, which in turn elicits intrinsic work motivation and job satisfaction. Autonomous jobs are expected to encourage higher levels of job satisfaction than controlled jobs because JA makes employees feel self-determined and free from external controls or constraints (Naqvi et al., 2013). As a job resource, JA can contribute to positive work outcomes (Schaufeli & Taris, 2014). Consistent with the studies above, Morgeson and Humphrey (2006) have pointed out that JA is an important contextual resource to positively influence employees’ attitudes and behaviours. Thus, the current study expects a boosting effect of job autonomy on job satisfaction. 

Previous studies have mainly examined the moderating role of job resources on the relationship between job demands and work outcomes (Xanthopoulou et al., 2007; Tadić et al., 2015; Shantz & Alfes, 2015; Mayende & Musenze, 2018). Hakanen et al. (2005) also suggested that resources like job autonomy could moderate the impact of job demands for the majority of jobs. Prior researches have endeavoured to elucidate the degree to which job resources, such as job autonomy, act as a buffer in the association between variables, namely the relationship between job demands and work outcomes. Nevertheless, it is noteworthy that scant studies have explored the moderating effect of job resources in relationships between positive variables. Dominguez et al. (2020) have identified a need for additional research on the moderating influence of job resources in the relationships among positive variables. In response to this call, Wan and Duffy (2022) have investigated the moderating effect of JA on the association between decent work and job satisfaction, discovering that JA can strengthen the positive link between these two variables. According to Dominguez et al. (2020), promoting elements of intrinsic motivation can enhance the relationship between intrinsic motivational resources and positive work attitudes, including work engagement and job satisfaction. Therefore, it is rational to infer that JA, as an intrinsic motivational organizational resource, may boost the effects of other job resources such as PC, PP and AC on job satisfaction. Additionally, this boost effect may stem from the accumulation of job resources, as proposed by Hobfoll (2011), wherein workers who have greater autonomy experience a stronger motivational effect from job resources compared to those with lower autonomy levels. In high-autonomy jobs, young individuals who have greater decision-making power and place a premium on career development and performance-based pay are given greater freedom to demonstrate their abilities and receive corresponding incentives. This, in turn, leads to increased job satisfaction. Moreover, employees who have high levels of job autonomy feel a stronger sense of connection to their organization when they are allowed to perform their duties in a way that aligns with their personal values and goals. This sense of connection may ultimately result in higher job satisfaction. The younger workforce embraces independence at the workplace and evaluates a job based on the salary, opportunities for autonomy, and the use of their talents at work (Lee et al., 2017). Employees would be intrinsically positive about their jobs when they are provided with more opportunities for autonomy (Fried & Ferris, 1987). Hackman (1980) identified JA as a highly desirable attribute of a job and noted its close relationships with job motivation and job attitudes. Li et al. (2021) postulated the direct relationship between JA and career self-management in regards to employees’ career growth and success. Wan and Duffy (2022) demonstrated the motivating effect of JA on employees and the positive influence of JA on JS. Apart from its motivating effect on JS directly, JA reflects an environment that is not restricted by formal procedures (Meyer et al., 2010). According to Chang et al. (2015), employees who perceive that their supervisors support autonomy tend to be more satisfied with their jobs and affectively committed to their organisation. Empowering leadership, which offers autonomy and developmental support, potentially contributes favourable influence on employees’ AC (Kim & Beehr, 2020). Haar and Spell (2009) highlighted the influence of different levels of JA on the perceived level of distributive fairness and its relationship to JS. Therefore, it is plausible that different levels of JA influence the influence of PC, PP, and AC on JS. As JA influences employees’ work attitudes, it serves as a crucial boundary condition for the relationships of PP, PC, and AC with JS. Thus, the current study proposed the following hypotheses for testing:

H8: JA positively moderates the relationship between PC and JS.

H9: JA positively moderates the relationship between PP and JS.

H10: JA positively moderates the relationship between AC and JS.

Comment 3: H8, H9, and H10 were rejected. The author needs to explain in detail in the conclusion of the manuscript why the author chose this moderate variable and whether the moderating effect is not valid can be supported by other literature. Otherwise, the selection of this variable would be meaningless.

Author(s) Reply: Thank you for your constructive comments and suggestions, Prof. Amendment highlighted (added – BLUE; removed RED) below:

Dominguez et al. (2020) have highlighted the necessity for additional research on the moderating impact of job resources in the relationships between positive variables. While previous studies have primarily concentrated on the buffering effects of job resources on job demands, little attention has been given to the moderating role of job resources. Consequently, there is an urgent need for more extensive research to gain a more comprehensive understanding of this complex interplay. Responding to this call, besides that, this study empirically proved the insignificant moderating effect of JA on the relationships of perceived career development opportunities (H8), perceived pay for performance (H9), and affective organizational commitment (H10) with job satisfaction. In other words, young employees take into account the importance of JA at the workplace, but different levels of job autonomy do not affect the influence of these factors on job satisfaction. This finding contrasted with previous studies, such as Iplik (2014), which showed that job autonomy moderates the relationship between employees' perceptions and job satisfaction. Similarly, Wan and Duffy (2022) found that job autonomy moderated the relationship between job resources, such as decent work, and job satisfaction. It is possible that different demographic groups could yield different results, and cultural or contextual differences may explain the current findings. Cultural differences can influence how individuals perceive and respond to job resources, such as job autonomy, which could affect the moderating role of job autonomy in the relationship between other job resources and job satisfaction. In contrast to countries that prioritize individualism, Chinese culture emphasizes group harmony and collectivism, which could lead to greater acceptance of limited job autonomy and a focus on fulfilling assigned responsibilities (Hong et al., 2018). Therefore, future research is encouraged to explore the moderating effect of job autonomy between positive variables in diverse contexts. 

REVIEWER #2: 

Dear Author

Thank you for sending your paper "Career Development Opportunity, Remuneration, and Organisational Commitment on Job Satisfaction and Turnover Intention: Moderating Effect of Job Autonomy" to this journal "PLOS One". I read your paper and gave my concern down here:

Comment 4: I suppose that the topic is too long and can be shorten.

Author(s) Reply: Thank you for your constructive comments and suggestions, Prof. Amendment highlighted (added – BLUE; removed RED) below:

INTRODUCTION

It has become increasingly crucial for companies to enhance their work efficiency due to the intense market competition. The workforce has substantial influence on the organisational or business success, which explains the significant need for companies to optimise their workforce, particularly the younger workforce considering their high turnover rate and independent nature (Ramlah et al., 2021). With the constant development of the economy and the increasing competition in the labour market, an increasing number of young employees are contemplating job-hopping or leaving their current company. The higher the turnover intention, the higher the likelihood of quitting the job (Jacobs & Roodt, 2008). Belete (2018) described turnover intention as the intention to leave the workplace. The management must have good understanding of the factors that contribute to turnover intention considering how a higher turnover rate would affect the organisational morale, reduce the sense of identity within the company (Joe et al., 2018), and yield negative financial impact since the companies have to allocate resources for recruitment and training and experience brain drain and lower productivity (Belete, 2018).

The older workforce possesses more traditional values, such as contributing to the country and supporting the family, but the younger workforce emphasises more worldly values and embraces self-development as the key purpose of working (Zhao, 2018). In general, the younger workforce demonstrates unique characteristics at work. They have different expectations at work and expect higher expectations on perceived career development opportunity (PC) and perceived pay for performance (PP) than the older workforce (Magni & Manzoni, 2020). Besides that, prior studies demonstrated lower affective organisational commitment (AC) among young employees than older employees (Singh & Gupta, 2015; Salminen & Miettinen, 2019). Adding to that, the younger workforce highly values job autonomy (JA) at the workplace. Morgeson and Humphrey (2006) defined JA as the extent of freedom and independence allowed for employees to arrange their work schedule, make decisions, and select the method of performing tasks at the workplace. The younger workforce is more independent in nature (Ramlah et al., 2021) and may be less loyal towards the company (Singh & Gupta, 2015). They are not willing to obey orders and instructions, as they prefer to work freely and independently (Lee et al., 2017). Moreover, the younger workforce emphasises how they feel, which makes them highly value job satisfaction (JS) at the workplace (Yi et al., 2010; Zhao, 2018). In short, unlike the older workforce, the younger workforce has higher expectations on PC and PP but lower expectations on AC compared with older workforce and display higher concern for JA and JS. 

Prior studies examined the influence of these unique attributes of young workforce on turnover intention (TI) (Lee et al., 2017; Belete, 2018; Alam & Asim, 2019; Wan & Duffy, 2022). However, none of these studies examined these relationships across different generations. As a result, the significance of these attributes as the main predictors of TI among the young workforce has remained unclear. Prior studies mainly focused on the direct influence of PC, PP, and AC on TI, respectively. Only a few studies justified PC, PP, and AC as predictors of TI. Besides that, the relationships of JA and JS with TI have remained underexplored, which were addressed in the current study. These young employees are the future workforce and soon replace the current management. Retaining these young employees can minimise the costs of recruiting and training new talents. Additionally, prior studies have mainly focused on the direct influence of PC, PP, and AC on TI and largely neglected the mechanism of how PC, PP and AC affect TI. Drawing upon the equity theory (Adams, 1965), which proposes that employees' satisfaction levels are contingent upon their perceptions of fairness regarding the relationship between their inputs and outputs, previous study has pointed out the mediating role of job satisfaction (JS) between the relationship between employee perceptions of organizational justice and employee outcomes (Mashi, 2018). Hence, the present study seeks to investigate the mechanism of how PC, PP and AC affect TI by focusing on JS.

Furthermore, limited research has explored whether there are certain conditions under which the effects of PC, PP and AC on job satisfaction and turnover intention most likely emerges. Examining these unexplored boundary conditions is critical, as it may offer valuable insights that the effects of PC, PP, and AC should not be taken for granted. According to Fried and Ferris (1987), when employees perceive more autonomous opportunities provided by the organization, they will have internal positive feelings toward their work. Job autonomy (JA) might be an important contextual resource to influence employees’ attitudes and behaviours (Morgeson & Humphrey, 2006). Hence, this study explores the role of JA within the organization as a boundary condition to the relationships of PC, PP and AC with job satisfaction and turnover intention. Overall, to address these gaps in the literature, the current study seeks to investigate whether PC, PP, and AC indirectly affect turnover intention by influencing job satisfaction under different levels of job autonomy.

Focusing on the young workforce in China, the current study quantitatively assessed the influence of PC, PP, and AC on JS and TI, as well as the moderating effect of JA. In particular, this study addressed the following research questions: (a) Does PC influence TI directly or indirectly under different levels of JA? (b) Does PP influence TI directly or indirectly under different levels of JA? And (c) Does AC influence TI directly or indirectly under different levels of JA? 

Comment 5: The research problem can be made crystal clear. Moreover, the very first sentence of the introduction section started with "it"! What does it mean? 'It' should be clear.

Author(s) Reply: Thank you for your constructive comments and suggestions, Prof. Amendment highlighted (added – BLUE; removed RED) below:

INTRODUCTION

It has become increasingly crucial for companies to enhance their work efficiency due to the intense market competition. The workforce has substantial influence on the organisational or business success, which explains the significant need for companies to optimise their workforce, particularly the younger workforce considering their high turnover rate and independent nature (Ramlah et al., 2021). With the constant development of the economy and the increasing competition in the labour market, an increasing number of young employees are contemplating job-hopping or leaving their current company. The higher the turnover intention, the higher the likelihood of quitting the job (Jacobs & Roodt, 2008). Belete (2018) described turnover intention as the intention to leave the workplace. The management must have good understanding of the factors that contribute to turnover intention considering how a higher turnover rate would affect the organisational morale, reduce the sense of identity within the company (Joe et al., 2018), and yield negative financial impact since the companies have to allocate resources for recruitment and training and experience brain drain and lower productivity (Belete, 2018).

…….

Prior studies examined the influence of these unique attributes of young workforce on turnover intention (TI) (Lee et al., 2017; Belete, 2018; Alam & Asim, 2019; Wan & Duffy, 2022). However, none of these studies examined these relationships across different generations. As a result, the significance of these attributes as the main predictors of TI among the young workforce has remained unclear. Prior studies mainly focused on the direct influence of PC, PP, and AC on TI, respectively. Only a few studies justified PC, PP, and AC as predictors of TI. Besides that, the relationships of JA and JS with TI have remained underexplored, which were addressed in the current study. These young employees are the future workforce and soon replace the current management. Retaining these young employees can minimise the costs of recruiting and training new talents. Additionally, prior studies have mainly focused on the direct influence of PC, PP, and AC on TI and largely neglected the mechanism of how PC, PP and AC affect TI. Drawing upon the equity theory (Adams, 1965), which proposes that employees' satisfaction levels are contingent upon their perceptions of fairness regarding the relationship between their inputs and outputs, previous study has pointed out the mediating role of job satisfaction (JS) between the relationship between employee perceptions of organizational justice and employee outcomes (Mashi, 2018). Hence, the present study seeks to investigate the mechanism of how PC, PP and AC affect TI by focusing on JS.

Furthermore, limited research has explored whether there are certain conditions under which the effects of PC, PP and AC on job satisfaction and turnover intention most likely emerges. Examining these unexplored boundary conditions is critical, as it may offer valuable insights that the effects of PC, PP, and AC should not be taken for granted. According to Fried and Ferris (1987), when employees perceive more autonomous opportunities provided by the organization, they will have internal positive feelings toward their work. Job autonomy (JA) might be an important contextual resource to influence employees’ attitudes and behaviours (Morgeson & Humphrey, 2006). Hence, this study explores the role of JA within the organization as a boundary condition to the relationships of PC, PP and AC with job satisfaction and turnover intention. Overall, to address these gaps in the literature, the current study seeks to investigate whether PC, PP, and AC indirectly affect turnover intention by influencing job satisfaction under different levels of job autonomy.

Focusing on the young workforce in China, the current study quantitatively assessed the influence of PC, PP, and AC on JS and TI, as well as the moderating effect of JA. In particular, this study addressed the following research questions: (a) Does PC influence TI directly or indirectly under different levels of JA? (b) Does PP influence TI directly or indirectly under different levels of JA? And (c) Does AC influence TI directly or indirectly under different levels of JA?

Comment 6: The hypothesis on moderating effect needs further improvement.

Author(s) Reply: Thank you for your constructive comments and suggestions, Prof. Amendment highlighted (added – BLUE; removed RED) below:

Job Autonomy

Spector (1986) described JA as the extent to which employees can make their own decisions based on their judgment and preferences on how they execute the assigned tasks. Specifically, when employees have autonomy at work, they can complete tasks in a free manner according to their judgment and preferences. The job characteristics model (Hackman & Oldham, 1975) proposed that core job characteristics including autonomy generate a sense of responsibility for outcomes, meaningfulness, and knowledge of results, which in turn elicits intrinsic work motivation and job satisfaction. Autonomous jobs are expected to encourage higher levels of job satisfaction than controlled jobs because JA makes employees feel self-determined and free from external controls or constraints (Naqvi et al., 2013). As a job resource, JA can contribute to positive work outcomes (Schaufeli & Taris, 2014). Consistent with the studies above, Morgeson and Humphrey (2006) have pointed out that JA is an important contextual resource to positively influence employees’ attitudes and behaviours. Thus, the current study expects a boosting effect of job autonomy on job satisfaction. 

Previous studies have mainly examined the moderating role of job resources on the relationship between job demands and work outcomes (Xanthopoulou et al., 2007; Tadić et al., 2015; Shantz & Alfes, 2015; Mayende & Musenze, 2018). Hakanen et al. (2005) also suggested that resources like job autonomy could moderate the impact of job demands for the majority of jobs. Prior researches have endeavoured to elucidate the degree to which job resources, such as job autonomy, act as a buffer in the association between variables, namely the relationship between job demands and work outcomes. Nevertheless, it is noteworthy that scant studies have explored the moderating effect of job resources in relationships between positive variables. Dominguez et al. (2020) have identified a need for additional research on the moderating influence of job resources in the relationships among positive variables. In response to this call, Wan and Duffy (2022) have investigated the moderating effect of JA on the association between decent work and job satisfaction, discovering that JA can strengthen the positive link between these two variables. According to Dominguez et al. (2020), promoting elements of intrinsic motivation can enhance the relationship between intrinsic motivational resources and positive work attitudes, including work engagement and job satisfaction. Therefore, it is rational to infer that JA, as an intrinsic motivational organizational resource, may boost the effects of other job resources such as PC, PP and AC on job satisfaction. Additionally, this boost effect may stem from the accumulation of job resources, as proposed by Hobfoll (2011), wherein workers who have greater autonomy experience a stronger motivational effect from job resources compared to those with lower autonomy levels. In high-autonomy jobs, young individuals who have greater decision-making power and place a premium on career development and performance-based pay are given greater freedom to demonstrate their abilities and receive corresponding incentives. This, in turn, leads to increased job satisfaction. Moreover, employees who have high levels of job autonomy feel a stronger sense of connection to their organization when they are allowed to perform their duties in a way that aligns with their personal values and goals. This sense of connection may ultimately result in higher job satisfaction. The younger workforce embraces independence at the workplace and evaluates a job based on the salary, opportunities for autonomy, and the use of their talents at work (Lee et al., 2017). Employees would be intrinsically positive about their jobs when they are provided with more opportunities for autonomy (Fried & Ferris, 1987). Hackman (1980) identified JA as a highly desirable attribute of a job and noted its close relationships with job motivation and job attitudes. Li et al. (2021) postulated the direct relationship between JA and career self-management in regards to employees’ career growth and success. Wan and Duffy (2022) demonstrated the motivating effect of JA on employees and the positive influence of JA on JS. Apart from its motivating effect on JS directly, JA reflects an environment that is not restricted by formal procedures (Meyer et al., 2010). According to Chang et al. (2015), employees who perceive that their supervisors support autonomy tend to be more satisfied with their jobs and affectively committed to their organisation. Empowering leadership, which offers autonomy and developmental support, potentially contributes favourable influence on employees’ AC (Kim & Beehr, 2020). Haar and Spell (2009) highlighted the influence of different levels of JA on the perceived level of distributive fairness and its relationship to JS. Therefore, it is plausible that different levels of JA influence the influence of PC, PP, and AC on JS. As JA influences employees’ work attitudes, it serves as a crucial boundary condition for the relationships of PP, PC, and AC with JS. Thus, the current study proposed the following hypotheses for testing:

H8: JA positively moderates the relationship between PC and JS.

H9: JA positively moderates the relationship between PP and JS.

H10: JA positively moderates the relationship between AC and JS.

Comment 7: The methodology section seems alright.

Author(s) Reply: Thank you for your constructive comments and suggestions, Prof. Amendment highlighted (added – BLUE; removed RED) below:

METHODOLOGY

Data Collection

The young workforce in China generally has lower organisational loyalty and is associated with higher turnover rate (Warner & Zhu, 2018; Fang et al., 2020). Therefore, it was deemed significant for the current study to assess the influence of PC, PP, and AC on JS and TI, as well as the moderating effect of JA among the young employees in China. Referring to the China State Council (2022), youths are of those between 14 and 35 years old. Considering the context of the current study, working individuals of between 18 and 35 years old in China were targeted.

This research employed a quantitative approach to examine the relationship between variables. G*Power 3.1 was used to calculate the minimum sample size required to achieve the target analysis level and the desired minimum sample size with a power of 0.95 and an effect size of 0.15, taking into account the five predictors in this study. The output result indicates that at least 138 valid samples are required (Faul et al., 2007). However, for partial least squares-structural equation modeling analysis, it is recommended to use a minimum of 200 samples (Hair, 2017). Online data collection was conducted between July 1, 2022 and August 8, 2022. The online survey was administered through WJX (http://www.wjx.cn/), a functional and user-friendly tool. In order to ensure that all participants were young Chinese adults who were employed and aligned with the background of this study, judgmental questions were used for exclusion purposes. Prior to the commencement of the formal questionnaire, all participants signed an informed consent form, in which they were informed of the purpose of the data collection, the final destination of the data, and their right to withdraw from participation at any time. Using the G-Power 3.1, this study estimated the minimum sample size required. With the power of 0.95, effect size of 0.15, and six predictors, this study was expected to gather at least 146 effective samples (Faul et al., 2007). According to Hair (2017), a sample size of at least 200 is recommended for partial least squares-structural equation modelling (PLS-SEM). Therefore, this study opted for a sample size of at least 500 respondents to reduce the effects of low sample size. Convenience sampling strategy was employed in this study. An online survey was conducted via WJX (http://www.wjx.cn/), which is a functional and user-friendly tool. The online survey included several qualifying questions to filter potential respondents. At the end of the data collection, this study successfully obtained 532 valid questionnaire sets.

Comment 8: I missed the gravity of the discussion section in this manuscript. Would you add discussion section here?

Author(s) Reply: Thank you for your constructive comments and suggestions, Prof. Amendment highlighted (added – BLUE; removed RED) below:

DISCUSSION AND IMPLICATIONS

This cross-sectional study exclusively focused on turnover intention of young employees in China. Retaining young employees is undoubtedly a major challenge for organisations today. Therefore, it is crucial for organisations to understand how attached their employees are to the workplace. Focusing on that, the current study presented empirical evidence on factors that influence TI. Firstly, this study empirically proved the direct significant influence of perceived career development opportunities on job satisfaction (H1) and turnover intention (H2). Stahl et al. (2009) and Ohunakin et al. (2018) presented similar findings on the substantial influence of perceived career development opportunities on turnover intention. Likewise, Price and Reichert (2017) identified perceived career development opportunities as one of the factors that can enhance job satisfaction. In other words, PC significantly influences JS and TI. This study also empirically demonstrated the significant and positive influence of perceived pay for performance on job satisfaction (H3). This finding was consistent with the studies conducted by Ren et al. (2017) and Bae (2021), which have reported the significant and positive relationship between perceived pay for performance and job satisfaction. This finding highlighted the importance of perceived pay for performance in motivating young Chinese employees and improving job satisfaction. Additionally, the study discovered that perceived pay for performance had a negative impact on turnover intention, providing support for H4. This finding aligned with the assertions by Kuvaas et al. (2016) and Hur and Ha (2019). They pointed out performance-based pay had a negative impact on turnover intention. In other words, among young Chinese employees, the recognition and appropriate compensation of their efforts and contributions may lead to a decreased likelihood of contemplating job departure., which were found to be only partially consistent with the results reported by Ren et al. (2017). The prior study reported the significant and positive relationship between PP and JS but insignificant relationship between PP and TI. Nonetheless, the current study hypothesised the significant and negative influence of PP on TI. Based on the results, H4 was supported. 

Besides that, this study empirically demonstrated the positive influence of affective organizational commitment on job satisfaction. Thus H5 was supported. This finding was consistent with several prior studies (Ozturk et al., 2014; Yang et al., 2019; Cao et al., 2020). In other words, employees who have a strong emotional attachment to their organization are more likely to experience job satisfaction. Furthermore, the current study shed light on the relationship between affective organizational commitment and turnover intention in the context of young Chinese workers. Specifically, the study revealed that employees' affective organizational commitment negatively influenced their turnover intentions (H6). Gara Bach Ouerdian et al. (2021) and Karatepe (2022) presented similar findings on the substantial influence of affective organizational commitment on turnover intention. The current study adds to the existing literature on the importance of job resources such as affective organizational commitment in reducing employee turnover intention, which has implications for organizations seeking to improve their retention rate. The obtained results of this study also confirmed JS as a significant determinant of TI (H7), which supported the findings of several prior studies (Alam & Asim, 2019; Wan & Duffy, 2022). In other words, the significance of JS as a predictor of TI should not be overlooked.

Besides that, this study empirically demonstrated the positive influence of AC on JS (H5) and the negative influence of AC on TI (H6), which supported the findings of several prior studies (Yang et al., 2019; Cao et al., 2020; Obeng et al., 2021). In other words, young employees with AC display lower TI and tend to express higher JS. However, there have varied findings on the direction of the relationship between AC and JS. For instance, Chordiya et al. (2017) identified JS as the most important antecedent of AC, while Berta et al. (2018) highlighted ambiguous direction of the positive relationship between AC and JS.

 Dominguez et al. (2020) have highlighted the necessity for additional research on the moderating impact of job resources in the relationships between positive variables. While previous studies have primarily concentrated on the buffering effects of job resources on job demands, little attention has been given to the moderating role of job resources. Consequently, there is an urgent need for more extensive research to gain a more comprehensive understanding of this complex interplay. Responding to this call, Besides that, this study empirically proved the insignificant moderating effect of JA on the relationships of perceived career development opportunities (H8), perceived pay for performance (H9), and affective organizational commitment (H10) with job satisfaction. In other words, young employees take into account the importance of JA at the workplace, but different levels of job autonomy do not affect the influence of these factors on job satisfaction. This finding contrasted with previous studies, such as Iplik (2014), which showed that job autonomy moderates the relationship between employees' perceptions and job satisfaction. Similarly, Wan and Duffy (2022) found that job autonomy moderated the relationship between job resources, such as decent work, and job satisfaction. It is possible that different demographic groups could yield different results, and cultural or contextual differences may explain the current findings. Cultural differences can influence how individuals perceive and respond to job resources, such as job autonomy, which could affect the moderating role of job autonomy in the relationship between other job resources and job satisfaction. In contrast to countries that prioritize individualism, Chinese culture emphasizes group harmony and collectivism, which could lead to greater acceptance of limited job autonomy and a focus on fulfilling assigned responsibilities (Hong et al., 2018). Therefore, future research is encouraged to explore the moderating effect of job autonomy between positive variables in diverse contexts. 

Theoretical Implications

This paper supports the call for research into the impact of employees' interpretations of HRM practices on the effectiveness of the HRM system (Kehoe & Wright, 2013; Ren et al., 2017). This study presented significant theoretical implications that would be of much interest for future research on TI, particularly among young employees. This study addressed the call of prior studies (Brokmeier et al., 2022) on the need for more in-depth insights on how job characteristics influence the formation of work attitudes among employees. This study presented empirical evidence on factors that can reduce young employees’ TI in relation to the unique attributes of this young workforce. In particular, the significance of PC, PP, and AC in influencing TI was highlighted. This study demonstrated the importance of considering the work characteristics of young employees when it comes to exploring the formation of their JS and TI. This study also presented unexpected findings on how JA does not moderate the relationships of PC, PP, and AC with JS despite the importance of JA for young employees at the workplace.

Practical Implications

This study also presented several practical implications that would be of much interest for various organisations and their management. First, this study clearly demonstrated the importance of enhancing PC, PP, and AC for employees to have higher JS and subsequently, TI. The results of this study hold significant practical implications for organizations, particularly those operating in China, seeking to reduce turnover intention among young employees. To attain this objective, organizations must concentrate on augmenting perceived career development opportunities, perceived pay for performance, and affective organizational commitment. To improve employee perceptions of career development opportunities, organizations can facilitate training and development programs designed to enhance employees' skills and competencies, provide opportunities for job rotation, and mentorship programs. Furthermore, managers can implement performance-based pay systems, furnish regular feedback and recognition to employees, and cultivate a culture of accountability and transparency. Such measures will elevate employees' perception of fairness in the pay structure and stimulate their motivation to perform well. 

This study also demonstrated AC as the most important element with the highest overall effects on TI among young employees in China. However, it is the least-valued element by the management. Therefore, organizations should prioritize the development of strategies and initiatives that enhance employees' affective commitment to the organization. Organizations can create a positive work environment that fosters trust, openness, and communication (Men & Yue, 2019). This can be achieved by promoting a strong organizational culture, providing opportunities for employee involvement and participation in decision-making, and recognizing and rewarding employees' contributions. This study prompted the crucial need for the management to identify strategic ways that can enhance PP, PC, and AC in order to reduce TI among young employees. 

Wish you all the best.

6. PLOS authors have the option to publish the peer review history of their article (what does this mean?). If published, this will include your full peer review and any attached files.

Do you want your identity to be public for this peer review? For information about this choice, including consent withdrawal, please see our Privacy Policy.

Reviewer #1: No

Reviewer #2: Yes: Md. Aftab Uddin

---

## [Decision Letter · Decision Letter 1]

28 Mar 2023

PONE-D-23-03002R1Career Development Opportunity, Remuneration, and Organisational Commitment on Job Satisfaction and Turnover Intention: Moderating Effect of Job AutonomyPLOS ONE

Dear Dr. Al Mamun,

Thank you for submitting your manuscript to PLOS ONE. After careful consideration, we feel that it has merit but does not fully meet PLOS ONE’s publication criteria as it currently stands. Therefore, we invite you to submit a revised version of the manuscript that addresses the points raised during the review process.

We look forward to receiving your revised manuscript.

Kind regards,

Bo Pu, Ph.D.

Academic Editor

PLOS ONE

Journal Requirements:

Reviewers' comments:

Reviewer's Responses to Questions

**Comments to the Author**

1. If the authors have adequately addressed your comments raised in a previous round of review and you feel that this manuscript is now acceptable for publication, you may indicate that here to bypass the “Comments to the Author” section, enter your conflict of interest statement in the “Confidential to Editor” section, and submit your "Accept" recommendation.

Reviewer #1: All comments have been addressed

Reviewer #2: All comments have been addressed

2. Is the manuscript technically sound, and do the data support the conclusions?

Reviewer #1: Yes

Reviewer #2: Partly

3. Has the statistical analysis been performed appropriately and rigorously? 

Reviewer #1: Yes

Reviewer #2: Yes

4. Have the authors made all data underlying the findings in their manuscript fully available?

Reviewer #1: Yes

Reviewer #2: No

5. Is the manuscript presented in an intelligible fashion and written in standard English?

Reviewer #1: Yes

Reviewer #2: Yes

6. Review Comments to the Author

Reviewer #1: (No Response)

Reviewer #2: Dear Author

Thank you for submitting your paper "Career Development Opportunity, Remuneration, and Organisational Commitment on Job Satisfaction and Turnover Intention: Moderating Effect of Job Autonomy" to this esteemed journal "PlosOne". I read the manuscript carefully and gave my concern down here:

1. The title might be changed.

2. Abstract seems to rewrite again. It seems that the authors started with the objective of the study which demand a background of the study.

3. Keywords were chosen from the title. The authors might rethink to use or copy exactly from the title.

4. The first sentence of the introduction needs citation.

5. What is the research problem of the study? To me, the research problem must come out from the first paragraph of the introduction section.

6. After complete reading of the introduction section, it appears to me that the authors failed to accurately show the research problem or the significance of the problem to study or research. Moreover, those citations that justified their are outdated which also challenged the originality of the study. I will recommend to accurately portray the problem with the latest citations.

7. I have found very little justification of using second theory "theory of socio-emotional selectivity" to grip the model. Thus, the author might reconsider to use it or else delete it.

8. The justification of hypotheses 1-7 is loosely given. I will suggest to strengthen those.

9. Methods and analyses seem alright.

10. Discussion and other sections seem alright.

Wish you all the best.

7. PLOS authors have the option to publish the peer review history of their article (what does this mean?). If published, this will include your full peer review and any attached files.

Reviewer #1: **Yes: **wei hu

Reviewer #2: **Yes: **Md. Aftab Uddin, PhD

While revising your submission, please upload your figure files to the Preflight Analysis and Conversion Engine (PACE) digital diagnostic tool, https://pacev2.apexcovantage.com/. PACE helps ensure that figures meet PLOS requirements. To use PACE, you must first register as a user. Registration is free. Then, login and navigate to the UPLOAD tab, where you will find detailed instructions on how to use the tool. If you encounter any issues or have any questions when using PACE, please email PLOS at figures@plos.org. Please note that Supporting Information files do not need this step.<quillbot-extension-portal></quillbot-extension-portal>

---

## [Author Response · Author response to Decision Letter 1]

12 Apr 2023

Comments to the Author

1. If the authors have adequately addressed your comments raised in a previous round of review and you feel that this manuscript is now acceptable for publication, you may indicate that here to bypass the “Comments to the Author” section, enter your conflict of interest statement in the “Confidential to Editor” section, and submit your "Accept" recommendation.

Reviewer #1: All comments have been addressed

Reviewer #2: All comments have been addressed

2. Is the manuscript technically sound, and do the data support the conclusions?

Reviewer #1: Yes

Reviewer #2: Partly

Author(s) Reply: Thank you for your feedback prof. Please check the reply to each specific comments posted in Section 6. Review Comments to the Author

3. Has the statistical analysis been performed appropriately and rigorously? 

Reviewer #1: Yes

Reviewer #2: Yes

4. Have the authors made all data underlying the findings in their manuscript fully available?

Reviewer #1: Yes

Reviewer #2: No

Author(s) Reply: Thank you for your feedback prof. Complete data submitted as supporting material together with the manuscript

5. Is the manuscript presented in an intelligible fashion and written in standard English?

Reviewer #1: Yes

Reviewer #2: Yes

6. Review Comments to the Author

Reviewer #1: (No Response)

Reviewer #2: Dear Author

Thank you for submitting your paper "Career Development Opportunity, Remuneration, and Organisational Commitment on Job Satisfaction and Turnover Intention: Moderating Effect of Job Autonomy" to this esteemed journal "PlosOne". I read the manuscript carefully and gave my concern down here:

Comment 1. The title might be changed.

Author(s) Reply: Thank you for your feedback Prof., all amendment highlighted BLUE

Career Development Opportunity, Remuneration, and Organisational Commitment on Job Satisfaction and Turnover Intention: Moderating Effect of Job Autonomy 

Envisaging the Job Satisfaction and Turnover Intention among the Young Workforce: Evidence from an Emerging Economy

Comment 2. Abstract seems to rewrite again. It seems that the authors started with the objective of the study which demand a background of the study.

Author(s) Reply: Thank you for your feedback Prof., all amendment highlighted BLUE

Abstract

As the economy evolves and markets change after Covid-19, demand and competition in the labour market increase in China, and employees become increasingly concerned about their career opportunities, pay, and organizational commitment. This category of factors is often considered a key predictor of turnover intentions and job satisfaction, and it is important that companies and management have a good understanding of the factors that contribute to job satisfaction and turnover intentions. The purpose of this study was to investigate the factors that influence employees' job satisfaction and turnover intention and to examine the moderating role of employees' job autonomy. This cross-sectional study aimed to quantitatively assess the influence of perceived career development opportunity, perceived pay for performance, and affective organisational commitment on job satisfaction and turnover intention, as well as the moderating effect of job autonomy. An online survey, which involved 532 young workforce in China, was conducted. All data were subjected to partial least squares-structural equation modelling (PLS-SEM). The obtained results demonstrated the direct influence of perceived career development, perceived pay for performance, and affective organisational commitment on turnover intention. These three constructs were also found to have indirect influence on turnover intention through job satisfaction. Meanwhile, the moderating effect of job autonomy on the hypothesised relationships was not statistically significant. This is one of the earliest studies conducted after the COVID-19 disruption, as working adults around the world are realising that they may need to rearrange their priorities and that there is more to life than work. This study presented significant theoretical contributions on turnover intention in relation to the unique attributes of young workforce. The obtained findings may also benefit managers in their efforts of understanding the turnover intention of the workforce and promoting empowerment practices. 

Comment 3. Keywords were chosen from the title. The authors might rethink to use or copy exactly from the title.

Author(s) Reply: Thank you for your feedback Prof., all amendment highlighted BLUE

Keywords: Career Development Opportunity; Pay for Performance; Organizational Commitment; Turnover Intention of Youth Employees; Job Satisfaction of Youth Employees; Job Autonomy; PLS-SEM

Comment 4. The first sentence of the introduction needs citation.

Author(s) Reply: Thank you for your feedback Prof., all amendment highlighted BLUE

With the constant development of the economy and the increasing competition in the labour market, an increasing number of young employees are contemplating job-hopping or leaving their current company(Fan & DeVaro, 2019; Wang et al., 2019; Miao et al., 2020).

Comment 5. What is the research problem of the study? To me, the research problem must come out from the first paragraph of the introduction section.

Author(s) Reply: Thank you for your feedback Prof., all amendment highlighted BLUE

The management must have good understanding of the factors that contribute to turnover intention considering how a higher turnover rate would affect the organisational morale, reduce the sense of identity within the company (Joe et al., 2018), and yield negative financial impact since the companies have to allocate resources for recruitment and training and experience brain drain and lower productivity (Belete, 2018). Therefore, how to reduce the turnover intention among the young workforce and improve employee job satisfaction is an area that most organizations need to consider.

 In addition, under the impact of Covid-19 on the economy and labor market, China's youth labor market is facing a serious social problem-high brain drain most young new employees will leave or jump jobs within 2-3 years, with most of them being fresh university graduates, which in turn brings some hidden problems such as social problems of jobless youth (Miao et al., 2020). Therefore, improving job satisfaction and reducing the turnover rate of young employees during their working life is an urgent issue that most Chinese companies and organizations need to address. Due to China's unique family planning policy, most young employees are now as independent and capable as only children and are more focused on their own interests and development, which in turn leads to a stronger attachment to salary and development opportunities due to both financial and family pressures (Miao et al., 2020; Wan & Duffy，2022). Therefore, understanding the perception and importance of salary and career development opportunities among young Chinese employees can effectively help companies and organizations find the right solution to retain this fresh and capable young workforce.

Comment 6. After complete reading of the introduction section, it appears to me that the authors failed to accurately show the research problem or the significance of the problem to study or research. Moreover, those citations that justified there are outdated which also challenged the originality of the study. I will recommend to accurately portray the problem with the latest citations.

Author(s) Reply: Thank you for your feedback Prof., all amendment highlighted BLUE

Currently, the young millennial workforce has become the largest group in the labor market, and how to retain these young employees has been an ongoing issue and challenge for most organizations and managers (Jaharuddin & Zainol, 2019; Chavadi et al., 2021). In the past, the older workforce possesses more traditional values, such as contributing to the country and supporting the family, but the younger workforce emphasises more worldly values and embraces self-development as the key purpose of working (Zhao, 2018). In general, the younger workforce demonstrates unique characteristics at work. They have different expectations at work and expect higher expectations on perceived career development opportunity (PC) and perceived pay for performance (PP) than the older workforce (Magni & Manzoni, 2020). Besides that, prior studies demonstrated lower affective organisational commitment (AC) among young employees than older employees (Singh & Gupta, 2015; Salminen & Miettinen, 2019). In short, unlike the older workforce, the younger workforce has higher expectations on PC and PP but lower expectations on AC. However, compared to middle-aged and older employees with more work experience and years of experience, the younger workforce focuses more on a number of unique attributes that in turn can have a significant impact on their job satisfaction and turnover intention, such as room for personal growth and development (Lee et al., 2017), pay level and task clarity (Alam & Asim, 2019), job variety and job autonomy (Wan & Duffy, 2022). Therefore, this study attempts to examine the mechanisms influencing young employees' job satisfaction and turnover intention, as well as the moderating role of job autonomy, by looking at some unique attributes, such as personal career development opportunities. 

Prior studies examined the influence of these unique attributes of young workforce on turnover intention (TI) (Lee et al., 2017; Belete, 2018; Alam & Asim, 2019; Wan & Duffy, 2022). However, none of these studies examined these relationships across different generations. As a result, the significance of these attributes as the main predictors of TI among the young workforce has remained unclear. Additionally, prior studies have mainly focused on the direct influence of PC, PP, and AC on TI and largely neglected the mechanism of how PC, PP and AC affect TI. Drawing upon the equity theory (Adams, 1965), which proposes that employees' satisfaction levels are contingent upon their perceptions of fairness regarding the relationship between their inputs and outputs, previous study has pointed out the mediating role of job satisfaction (JS) between the relationship between employee perceptions of organizational justice and employee outcomes (Mashi, 2018). Hence, the present study seeks to investigate the mechanism of how PC, PP and AC affect TI by focusing on JS. In addition, under the impact of Covid-19 on the economy and labor market, China's youth labor market is facing a serious social problem-high brain drain most young new employees will leave or jump jobs within 2-3 years, with most of them being fresh university graduates, which in turn brings some hidden problems such as social problems of jobless youth (Miao et al., 2020). Therefore, improving job satisfaction and reducing the turnover rate of young employees during their working life is an urgent issue that most Chinese companies and organizations need to address. Due to China's unique family planning policy, most young employees are now as independent and capable as only children and are more focused on their own interests and development, which in turn leads to a stronger attachment to salary and development opportunities due to both financial and family pressures (Miao et al., 2020; Wan & Duffy，2022). Therefore, understanding the perception and importance of salary and career development opportunities among young Chinese employees can effectively help companies and organizations find the right solution to retain this fresh and capable young workforce.

Furthermore, limited research has explored whether there are certain conditions under which the effects of PC, PP and AC on job satisfaction and turnover intention most likely emerges. Examining these unexplored boundary conditions is critical, as it may offer valuable insights that the effects of PC, PP, and AC should not be taken for granted. According to Fried and Ferris (1987), when employees perceive more autonomous opportunities provided by the organization, they will have internal positive feelings toward their work. Job autonomy (JA) might be an important contextual resource to influence employees’ attitudes and behaviours (Morgeson & Humphrey, 2006). Hence, this study explores the role of JA within the organization as a boundary condition to the relationships of PC, PP and AC with job satisfaction and turnover intention. Overall, to address these gaps in the literature, the current study seeks to investigate whether PC, PP, and AC indirectly affect turnover intention by influencing job satisfaction under different levels of job autonomy. Furthermore, young workers in the workplace usually focus on things that are related to their own interests, such as career development opportunities and salary (Magni & Manzoni, 2020; Chavadi et al., 2021). Young employees are ready to change jobs when the benefits and development space provided by the job do not meet their expectations and satisfy their needs (Miao et al., 2020). At the same time, the imbalance between job income and personal living expenses will also put more stress on them, which in turn will lead to higher intention of young employees to leave their jobs (Jaharuddin & Zainol, 2019). In addition, affective commitment is based on the emotional bond that employees build with the organization through positive work experiences, and some young employees' lack of work experience and skills, among other factors, may lead them to be reluctant to spend too much time building a strong emotional bond with the organization, which in turn leads to the idea of low cost of leaving and dissatisfaction with the job status quo (Zhou et al., 2020). Therefore, this study wanted to explore the perceptions and the level of importance of career development opportunities, pay for performance, and affective organizational commitment for young workers, and then discuss in depth whether these factors have some impact on the turnover intention and job satisfaction of young workers.

Comment 7. I have found very little justification of using second theory "theory of socio-emotional selectivity" to grip the model. Thus, the author might reconsider to use it or else delete it.

Author(s) Reply: Thank you for your feedback Prof., all amendment highlighted BLUE

Meanwhile, the theory of socio-emotional selectivity (Carstensen et al., 1999) suggests the influence of one’s perception of whether time is limited or unlimited on the formation of goals. In this case, younger individuals tend to perceive that time is unlimited. Therefore, they are more likely to form goals that can benefit their future, such as career- and knowledge-focused goals. On the other hand, older individuals tend to prioritise the present and take a closer approach to emotion-related goals, such as participating in activities that offer emotionally gratifying experiences, due to their perception of limited time (Zaniboni et al., 2013). Young employees tend to express higher satisfaction if they have the opportunities to improve themselves with valuable skills that can benefit their future work life (Loughlin & Barling, 2001). In short, it is highly plausible that younger employees prioritise development opportunities as they have the perception of unlimited time for future career growth, whereas older employees place lower emphasis on development opportunities due to their perception of limited time. the Job Demands-Resources (JD-R) Model was first introduced by Demerouti et al. (2001) and gained high popularity among researchers. The JD-R model posits that any job can be described by two sets of variables: job demands and job resources (Bakker et al., 2003). In particular, job resources are defined by Demerouti et al. (2001) as “those physical, social, or organizational aspects of the job that may serve any of the following functions: (a) facilitate achievement of work goals, (b) reduce job demands and their associated physiological and psychological costs, and (c) promote personal growth and development.” Hence, career development opportunities, pay for performance and affective organizational commitment can all be considered job resources, since job resources include any aspects of the job that can stimulate personal growth and development. Job resources can contribute to positive outcomes and also act as protective factors against negative outcomes, such as turnover intention (Schaufeli & Taris, 2014). In other words, employees who have access to more job resources are less likely to exhibit turnover intention.

Comment 8. The justification of hypotheses 1-7 is loosely given. I will suggest to strengthen those.

Author(s) Reply: Thank you for your feedback Prof., all amendment highlighted BLUE

Development of Hypotheses

Perceived Career Development Opportunity (PC)

Ayodele et al. (2020) defined PC as a formal organised plan that aligns with an employee’s career goals with respect to the organisational needs. PC refer to the employees' perceptions of the availability of work assignments and job opportunities that align with their career interests and goals within their current organization (Kraimer et al., 2011). Training for career advancement and growth can create the sense of being valued and appreciated for employees (Al Bastaki et al., 2021). Employees’ JS can be enhanced through professional development opportunities and flexible working hours (with more scheduling options) (Price & Reichert, 2017). This finding is corroborated by the JD-R model, which suggests that employees who have access to job resources tend to be more engaged and satisfied at work (Schaufeli & Taris, 2014). Meanwhile, Muleya et al. (2022) have underscored the impact of career development opportunity, regarded as one key job resource, on employees' attitudes in the workplace. It is reasonable to assume that when employees are provided with career development opportunities, they are more likely to exhibit positive work attitudes. Muleya et al. (2022) highlighted the positive influence of PC on employees’ attitudes and behaviours at the workplace. Meanwhile, Coetzee and Bester (2021) demonstrated the significant influence of JS and achievements of career goals on TI via PC. According to the theory of socio-emotional selectivity (Carstensen et al., 1999), younger employees place higher emphasis on PC than older employees, as they perceive that these development opportunities have substantial influence on their career growth. Additionally, Barhate and Dirani (2022) posit that the younger generation of workers require immediate rewards in the form of promotions and career advancement opportunities for a job well done. According to the equity theory (Adams, 1965), individuals may become demotivated and decrease their input, or seek change, if they perceive that their efforts are not being justly compensated. The advantages of career development and career planning within an organization include reducing employee turnover rates and increasing job satisfaction among employees (Dewi & Nurhayati, 2021). Huyghebaert et al. (2019) have examined the relationship between perceived career opportunities and turnover intentions in the nursing context, and their findings suggest that perceived career opportunities can prevent employees from intending to leave their jobs. Therefore, it is reasonable to infer that when younger employees believe that they are not being provided with adequate perceived career opportunities in their current workplace despite their hard work, they may feel a sense of injustice and consequently, become more likely to quit their jobs. In view of the above, the current study proposed the following hypotheses for testing:

H1: PC positively influences JS among young employees.

H2: PC negatively influences TI among young employees.

Perceived Pay for Performance (PP)

Gerhart and Fang (2015) defined pay for performance as a compensation programme that offers pay according to the performance in terms of outputs (e.g., sales volume) or behavioural evaluation. The correlation between pay growth and performance may not necessarily align with the correlation between employees' perceptions of performance-based pay and actual performance (Nyberg, 2010), which could affect employee attitudes. Consistent with the equity theory (Adams, 1965), employees’ perceptions of equity through the comparison of inputs and outputs affect their attitudes at the workplace. Ren et al. (2017) stressed the importance of assessing the influence of PP on employees’ attitudes at the workplace and identified PP and pay-level satisfaction as significant determinants of attitudes at the workplace (e.g., JS). Meanwhile, Kollmann et al. (2020) described the differences in JS between younger and older employees according to the monetary rewards, task contributions, and imbalances in the relationship between monetary rewards and task contributions. In a more recent study, Bae (2021) reported the significant and positive influence of perceived fairness of performance evaluation on pay satisfaction, organisational satisfaction, and JS. Younger workforce puts more emphasis on fairness and justice (Zhu et al., 2015). 

On the other hand, Kuvaas et al. (2016) demonstrated the significant influence of PP on TI via motivation as the mediator. Additionally, the JD-R model has posited that job resources (e.g. satisfying salary, appreciation and performance feedback) have motivational potential and may therefore lead to work engagement, which may result in positive organizational outcomes, including retention intentions (Schaufeli & Taris, 2014). The motivational impact of fair performance appraisal may be attenuated if employees perceive the performance appraisal process as lacking in fairness, validity, and reliability (Lee, 2019). As employees highly value distributive justice on PP, they tend to view their job continuity to be more predictable, controllable, and secure, which reduce the formation of TI (Hur & Ha, 2019). Hazeen Fathima and Umarani (2023) have provided empirical evidence that employees' perceived fairness of human resource practices, such as performance management and compensation, has a significant positive association with their intention to remain with the organization. In line with the equity theory, TI can be viewed as the resultant outcome of perceived inequity (Ngo-Henha, 2017). Considering that, creating a fair working environment serves as a key strategy for the management to reduce TI among the employees. Based on the findings of prior studies, the following hypotheses were proposed for testing:

H3: PP positively influences JS among young employees.

H4: PP negatively influences TI among young employees.

Affective Organisational Commitment (AC)

Meyer et al. (1990) and Buitendach and De Witte (2005) described AC as how employees are emotionally attached to, identify with, and involved with the organisation. AC is an individual attitudinal response that develops over time in response to an individual's employment experiences and beliefs about the work environment (Meyer et al., 1991), which in turn influences individuals’ work attitudes (Cao et al., 2020). Several studies have demonstrated that employees who exhibit affective commitment are intrinsically motivated, passionate about achieving organizational goals (Sharma & Dhar, 2016; Mercurio, 2015). Nauman et al. (2021) have pointed out the social exchange process by which training enhances the AC of employees which ultimately boosts employee’s job satisfaction. Apart from being the most consistent and significant predictor of TI, AC is the most widely acknowledged dimension of organisational commitment that consists of two other dimensions, namely continuance commitment and normative commitment (Yang et al., 2019). Prior studies examined the influence of AC on JS. For instance, Paik et al. (2007) revealed the positive influence of AC on JS and performance. Ozturk et al. (2014) similarly highlighted the significant influence of AC on JS. In a more recent study, Obeng et al. (2021) demonstrated the significant influence of both AC and JS on TI. Meanwhile, Cao et al. (2020) concluded the positive influence of AC on JS and its mediating effect on the relationship between work-family conflicts and JS. Meanwhile, Moreover, The impact of AC has been the focus of research in the field of organizational behaviour and has been proved to be intricately linked to both individual behaviour and organizational outcomes (Cao et al., 2020). Gara Bach Ouerdian et al. (2021) highlighted the significant and negative influence of AC on TI. Employees tend to develop emotional attachments with their organization when they experience positive interpersonal relationships with their colleagues and supervisors and perceive a congruence between their personal values and the organization's values (Ampofo & Karatepe, 2022; Ampofo, 2020), thus resulting in a decreased likelihood of turnover intentions among employees (Guzeller & Celiker, 2020). Hence, it is rational to infer that employees who possess AC demonstrate a heightened sense of belonging and connection to the organization's vision, coupled with a strong desire to remain with the organization. Such emotional attachment to the organization may also lead employees to subconsciously perceive their work as meaningful. Similar to other prior studies, Ampofo and Karatepe (2022) examined the mediating effect of AC on the relationship between on-the-job embeddedness and TI. Thus, the following hypotheses were tested in this study:

H5: AC positively influences JS among young employees. 

H6: AC negatively influences TI among young employees.

Job Satisfaction (JS)

Purani and Sahadev (2008) described JS as one’s sense of fulfilment gained from policies at the workplace, which are related to human resources, compensation, supervision, task clarity, and career development. The negative relationship between JS and TI has been demonstrated in numerous studies. Lin et al. (2022) have suggested that enhancing JS can yield multiple benefits for organizations, including decreased employee turnover and increased operational efficiencies, leading to cost savings. Bharadwaj et al. (2022) have proposed compelling evidence to support the notion that job satisfaction serves as a crucial antecedent to employee retention. This finding is supported by the JD-R model which posits the motivational process that job resources could stimulate positive work outcomes such as retention intention through positive work-related state (e.g. work engagement) or satisfaction of basic needs (Schaufeli & Taris, 2014). Additionally, some researchers have attempted to establish a relationship between JS and turnover intention. A potentially effective strategy for organizations to decrease employee turnover intention is to cultivate job satisfaction (Dodanwala et al., 2022). Alam and Asim (2019) have reported the significant and negative influence of different dimensions of job satisfaction (e.g. satisfaction with supervision, compensation, and career development) on TI. It is rational to infer employees who report higher levels of JS are more inclined to stay in their current workplace. Alam and Asim (2019) reported the significant and negative influence of satisfaction with supervision, compensation, and career development on TI. In other words, the study demonstrated the negative relationship between JS and TI. In a more recent study, Wan and Duffy (2022) identified several drivers, including JS, that influence the formation of TI. Thus, the current study tested the following hypothesis:

H7: JS negatively influences TI among young employees.

Comment 9. Methods and analyses seem alright.

Author(s) Reply: Thank you for your feedback Prof., all amendment highlighted BLUE

Comment 10. Discussion and other sections seem alright.

Author(s) Reply: Thank you for your feedback Prof., all amendment highlighted BLUE

We add some sentences to explain in detail in the discussion part why the moderating effect is not valid with the support by other literature. 

Tang and Vandenberghe (2019) found that affective organizational commitment may make employees feel driven by task autonomy, which can lead to job burnout, dissatisfaction, and intention to leave, and that employee job autonomy did not have a significant effect in this process; this is the same as the results of this study, where employee job autonomy did not moderate the effect between affective organizational commitment and employee job satisfaction. Wang et al. (2019) confirmed that employees' job satisfaction moderates the relationship between employees' job autonomy and turnover intention, and found a significant relationship between employees' salary factors (such as performance pay) and job satisfaction and it was not moderated by employees' job autonomy; this is consistent with the results of H9.

Wish you all the best.

7. PLOS authors have the option to publish the peer review history of their article (what does this mean?). If published, this will include your full peer review and any attached files.

Do you want your identity to be public for this peer review? For information about this choice, including consent withdrawal, please see our Privacy Policy.

Reviewer #1: Yes: wei hu

Reviewer #2: Yes: Md. Aftab Uddin, PhD

---

## [Decision Letter · Decision Letter 2]

4 Jun 2023

Envisaging the Job Satisfaction and Turnover Intention among the Young Workforce: Evidence from an Emerging Economy

PONE-D-23-03002R2

Dear Dr. Al Mamun,

We’re pleased to inform you that your manuscript has been judged scientifically suitable for publication and will be formally accepted for publication once it meets all outstanding technical requirements.

Kind regards,

Bo Pu, Ph.D.

Academic Editor

PLOS ONE

Additional Editor Comments (optional):

Reviewers' comments:

Reviewer's Responses to Questions

**Comments to the Author**

1. If the authors have adequately addressed your comments raised in a previous round of review and you feel that this manuscript is now acceptable for publication, you may indicate that here to bypass the “Comments to the Author” section, enter your conflict of interest statement in the “Confidential to Editor” section, and submit your "Accept" recommendation.

Reviewer #2: All comments have been addressed

2. Is the manuscript technically sound, and do the data support the conclusions?

Reviewer #2: Yes

3. Has the statistical analysis been performed appropriately and rigorously? 

Reviewer #2: Yes

4. Have the authors made all data underlying the findings in their manuscript fully available?

Reviewer #2: No

5. Is the manuscript presented in an intelligible fashion and written in standard English?

Reviewer #2: No

6. Review Comments to the Author

Reviewer #2: Dear Author

Thank you so much for your paper and I carefully read your paper. My concern is mentioned herewith:

I read your paper and it seems that it is well-written. However, my only concern is your paper's contribution. All the variables you have used are so old and there are mountainous literature on each of them. In this regard, would you please do a short literature review of last 3-5 years and include in appendix to confirm your originality. You might include a new paragraph or after discussion for attesting your originality based on this short literature review.

Thank you.

7. PLOS authors have the option to publish the peer review history of their article (what does this mean?). If published, this will include your full peer review and any attached files.

Reviewer #2: No

<quillbot-extension-portal></quillbot-extension-portal>

---

## [Editor Report · Acceptance letter]

9 Jun 2023

PONE-D-23-03002R2 

Envisaging the Job Satisfaction and Turnover Intention among the Young Workforce: Evidence from an Emerging Economy 

Dear Dr. Al Mamun:

I'm pleased to inform you that your manuscript has been deemed suitable for publication in PLOS ONE. Congratulations! Your manuscript is now with our production department. 

Kind regards, 

on behalf of

Dr. Bo Pu 

Academic Editor

PLOS ONE